



# Effective uncertainty quantification for multi-angle polarimetric aerosol remote sensing over ocean, Part 1: performance evaluation and speed improvement

Meng Gao[1,2], Kirk Knobelspiesse[1], Bryan A. Franz[1], Peng-Wang Zhai[3], Andrew M. Sayer[1,4], Amir Ibrahim[1], Brian Cairns[6], Otto Hasekamp[7], Yongxiang Hu[5], Vanderlei Martins[3,4], P. Jeremy Werdell[1], and Xiaoguang Xu[3,4]

[1]NASA Goddard Space Flight Center, Code 616, Greenbelt, Maryland 20771, USA
[2]Science Systems and Applications, Inc., Greenbelt, MD, USA
[3]JCET/Physics Department, University of Maryland, Baltimore County, Baltimore, MD 21250, USA
[4]Goddard Earth Sciences Technology and Research (GESTAR) II, University of Maryland, Baltimore County, Baltimore, MD 21250, USA
[5]MS 475 NASA Langley Research Center, Hampton, VA 23681-2199, USA
[6]NASA Goddard Institute for Space Studies, New York, NY 10025, USA
[7]Netherlands Institute for Space Research (SRON, NWO-I), Utrecht, The Netherlands

**Correspondence:** Meng Gao (meng.gao@nasa.gov)

**Abstract.** Multi-angle polarimetric (MAP) measurements can enable detailed characterization of aerosol microphysical and optical properties and improve atmospheric correction in ocean color remote sensing. Advanced retrieval algorithms have been developed to obtain multiple geophysical parameters in the atmosphere-ocean system. Theoretical pixel-wise retrieval uncertainties based on error propagation have been used to quantify retrieval performance and determine the quality of data

products. However, standard error propagation techniques in high-dimensional retrievals may not always represent true retrieval errors well due to issues such as local minima and nonlinearity of radiative transfer near the solution. In this work, we analyze these theoretical uncertainty estimates and validate them using a flexible Monte Carlo approach. The Fast Multi-Angular Polarimetric Ocean coLor (FastMAPOL) retrieval algorithm, based on several neural network forward models, is used to conduct the retrievals and uncertainty quantification on both synthetic HARP2 (Hyper-Angular Rainbow Polarimeter 2) and

AirHARP (airborne version of HARP2) datasets. In addition, for practical application of the technique to uncertainty evaluation in operational data processing, we use the automatic differentiation method to calculate derivatives analytically based on the neural network models. Both the speed and accuracy associated with uncertainty quantification for MAP retrievals are addressed in this study. Pixel-wise retrieval uncertainties are further evaluated for the real AirHARP field campaign data. The uncertainty quantification methods and results can be used to evaluate the quality of data products, and guide MAP algorithm

development for current and future satellite systems such as NASA's Plankton, Aerosol, Cloud, ocean Ecosystem (PACE) mission.



# 1 Introduction

Satellite remote sensing has revolutionized Earth observation capabilities and plays a significant role in studying atmosphere, ocean, and land systems. Remote sensing techniques have advanced rapidly to provide highly accurate geophysical prop-
erty retrievals by utilizing the rich information content of observations at multiple spectral bands, viewing angles and polarization states. Multi-angle polarimeters (MAPs) are particularly well-suited to characterize aerosol microphysical properties (Mishchenko and Travis, 1997; Chowdhary et al., 2001; Hasekamp and Landgraf, 2007; Knobelspiesse et al., 2012). Improved aerosol characterization helps reduce uncertainties in aerosol radiative forcing estimates and thereby advances our understanding of Earth's climate (Bender, 2020; Pörtner et al., In Press.). Furthermore, better quantification of the aerosol path radiance
in the atmosphere reduces error in the retrieval of spectral water-leaving radiances from ocean color remote sensing systems (Mobley et al., 2016; Mobley, 2022), which is important for the study of aquatic phytoplankton dynamics, marine ecosystems, and the global carbon cycle (Frouin et al., 2019; Groom et al., 2019).

Joint aerosol/ocean color retrieval algorithms have been developed for a variety of spaceborne and airborne MAPs such as the Polarization and Directionality of the Earth's Reflectances (POLDER) instruments (Hasekamp et al., 2011; Dubovik et al.,
2011, 2014; Li et al., 2019; Hasekamp et al., 2019b; Chen et al., 2020), the Airborne Multiangle SpectroPolarimetric Imager (AirMSPI) (Xu et al., 2016, 2019), the Spectro-Polarimeter for Planetary EXploration (SPEX) airborne (Fu and Hasekamp, 2018; Fu et al., 2020; Fan et al., 2019), SPEXone (spaceborne version of SPEX airborne) (Hasekamp et al., 2019b), the Research Scanning Polarimeter (RSP) (Chowdhary et al., 2005; Wu et al., 2015; Stamnes et al., 2018; Gao et al., 2018, 2019, 2020), the Directional Polarimetric Camera (DPC)/GaoFen-5 (Wang et al., 2014; Li et al., 2018), Airborne HyperAngular
Rainbow Polarimeter (AirHARP) (Puthukkudy et al., 2020; Gao et al., 2021a, b), and HARP2 (the spaceborne version of AirHARP) (Gao et al., 2021b). The algorithms typically follow iterative optimization approaches utilizing a vector radiative transfer forward model, and simultaneously retrieve a suite of geophysical parameters. A thorough review of MAP instruments and retrieval algorithms can be found in Dubovik et al. (2019).

Uncertainty quantification is an integral part of retrieval algorithm development. The uncertainties of the retrieved products
(hereafter 'retrieval uncertainties') are key to understanding retrieval performance, gauging whether the algorithm provides results of useful quality, and guiding where further efforts for improvement are best focused. In this study, we define retrieval error as the difference between the retrieval results and truth (whether synthetic data or external reference data), and retrieval uncertainty as the standard deviation ($1\sigma$) confidence interval around the retrieval solution (assuming a Gaussian distribution). Broadly, two methods are commonly used to determine retrieval uncertainties (see Sayer et al. (2020) for a review in the context
of aerosol remote sensing):

1. Propagated (hereafter 'theoretical') uncertainty: based on Bayesian theory, the uncertainty in observations and forward models as well as a priori assumption (hereafter 'input uncertainty model') can be mapped to the domain of retrieved parameters based on sensitivities derived from radiative transfer modeling (e.g. Rodgers (2000)). Pixel-wise uncertainties can be conveniently determined from an optimization algorithm based on its Jacobian matrix which represents the
measurement sensitivity with respect to the retrieval parameters.



However, theoretical uncertainties derived from these techniques often represent a best-case scenario as they rely on several assumptions (discussed by Povey and Grainger (2015)): a) the input uncertainty model is sufficient, b) the retrievals converge to their global minimum, c) the forward model is linear near the solution. Evaluating these assumptions for a given sensor and algorithm is therefore important. With MAPs, theoretical uncertainties have been widely used

for aerosol and cloud retrieval algorithms for sensors, such as POLDER (Hasekamp et al., 2011; Dubovik et al., 2011), RSP (Knobelspiesse et al., 2012), ground-based AERONET photo-polarimetric measurements (Xu and Wang, 2015; Xu et al., 2015), and general polarimetric instrument concept studies (Hasekamp and Landgraf, 2007; Knobelspiesse et al., 2012).

2. Truth-based (hereafter 'real' uncertainty): Retrieval errors are evaluated by comparing retrieval results with reference

data taken as a truth and used to draw general inference about retrieval uncertainties under various conditions. The 'real' uncertainty does not require the same assumptions as error propagation does but require the existence of 'truth' data of high and known confidence, which may be unavailable for some geophysical parameters. Additionally, the 'truth' data and matchup process have their own uncertainties which must be considered. In the absence of independent external truth, simulated retrievals are a useful tool. With MAPs, real uncertainties have been discussed for aerosols over ocean,

land, and cloud by comparing retrievals with synthetic data and in-situ measurements, such as for POLDER (Hasekamp et al., 2011; Dubovik et al., 2011; Chen et al., 2020), RSP (Chowdhary et al., 2012; Stamnes et al., 2018; Gao et al., 2019; Fu et al., 2020), AirMSPI (Xu et al., 2016), SPEX Airborne (Fu et al., 2020), SPEXone (Hasekamp et al., 2019a), AirHARP Puthukkudy:2020aa, Gao:2021aa, Gao:2021bb and HARP2 (Gao et al., 2021b).

In short, theoretical uncertainties provide pixel-wise estimates of performance for every parameter while real uncertainties

provide a more complete assessment of performance, but with limitations due to the availability of high-quality reference data. The two are a natural complement as ground-truth data or simulated retrievals provide an avenue to evaluate theoretical uncertainties in a statistical sense. A statistical (not one-to-one) comparison is necessary because a retrieval with associated uncertainty represents a range of plausible values of a geophysical quantity, whereas an individual reference truth has a definite value. Several approaches has been proposed to address the question whether the distribution of observed retrieval errors

is consistent with the distribution as expected from the theoretical uncertainty(Hasekamp and Landgraf, 2005; Sayer et al., 2020). For example, Hasekamp and Landgraf (2005) found the retrieval errors normalized by theoretical uncertainties from polarimetric retrievals can reproduce the general features of a Gaussian distribution, which was then used to discuss the impact of local minima and non-linearity around the truth. Sayer et al. (2020) illustrated a framework for aerosol retrievals based on normalized error distributions to quantitatively compare the real and theoretical uncertainties. Meanwhile, Monte Carlo

methods based on random sampling (Kalos and Whitlock., 2009), have been widely used to generate random error samples and used for analyzing their uncertainties (see Zhang (2021) for a survey) with applications to assess uncertainties of ocean bio-optical algorithms (McKinna et al., 2019). Monte Carlo methods are flexible and robust given sufficient sampling, but have not been well explored for MAP retrieval uncertainty studies.





In this paper, we discuss theoretical uncertainties from MAP retrievals over a coupled atmosphere and ocean system, and then propose a flexible framework to validate these theoretical uncertainties against real uncertainties. The following topics will be addressed in this work:

1. *Performance*: How well do theoretical uncertainties represent real retrieval uncertainties for both aerosol properties and the ocean color signal?

   This will be assessed not just for properties retrieved directly by the MAP, but also derived properties such as aerosol optical depth (AOD), single scattering albedo (SSA), and various aspects of the derived water-leaving signals. To quantify the performance in this study, random errors are sampled from theoretical pixel-wise uncertainties using a Monte Carlo method, and results are compared with the real errors.

2. *Speed*: How can uncertainty estimation be made sufficiently fast to be practical in operational data processing?

   Uncertainty evaluation often requires Jacobian matrix and derivative calculations, which can be computationally expensive. To achieve optimal speed within the framework of this work, all Jacobian matrix and derivatives are evaluated analytically using automatic differentiation based on neural networks.

3. *Input uncertainty model*: How representative is the algorithm's input uncertainty model?

   The input uncertainty model includes two main components: a) measurement uncertainties, which are mostly characterized by instrument calibration uncertainties, and b) forward model uncertainties, which refer to whether the forward model can sufficiently describe the measurements.

As Part 1 of a study series, this work focuses on the first two topics. The third topic has been partially addressed using an adaptive angular screening approach, described in (Gao et al., 2021b), to automatically remove MAP angles where the input uncertainty model is insufficient to describe forward model uncertainty due to contamination by cirrus clouds and other anomalies (Gao et al., 2021b). Note that noise correlation in the uncertainty model may impact retrieval results, though it is often ignored as assumed in this study (Knobelspiesse et al., 2012), and will be further addressed in the forthcoming Part 2 of this work. We study both theoretical and real uncertainties based on retrievals from synthetic AirHARP and HARP2 measurements, as well as AirHARP field measurement. This work provides a general approach to understand and evaluate pixel-wise uncertainties of high-dimensional retrieval problems, and can guide further uncertainty studies and algorithm development when more advanced instruments are available. Our primary focus is on these instruments due to HARP2's inclusion in the upcoming NASA's Plankton, Aerosol, Cloud, ocean Ecosystem (PACE) mission (Werdell et al., 2019), but the analysis is useful for future MAP missions, such as NASA's Multi-Angle Imager for Aerosols (MAIA) (Diner et al., 2018) and Atmosphere Observing System (AOS) missions (https://aos.gsfc.nasa.gov/), and the Multi-view Multi-channel Multi-polarization Imager (3MI) that will fly on ESA's MetOp-SG mission (Marbach et al., 2015). Section 2 of this paper describes the FastMAPOL retrieval algorithm used in the study; Section 3 discusses the methodology in the retrieval uncertainty evaluation; Section 4. quantified the performance of retrievals uncertainties based on synthetic AirHARP and HARP2 data ; Section 5. applied the pixel-wise uncertainties on the retrievals from AirHARP field measurements; and Section 6 provides discussions and conclusions.





## 2    FastMAPOL aerosol and ocean color retrieval algorithm

The FastMAPOL algorithm (Gao et al., 2021a) uses neural network forward models of a coupled atmosphere-ocean system, and has been used to perform retrievals on synthetic and observed AirHARP measurements (Gao et al., 2021a) and synthetic

HARP2 measurements (Gao et al., 2021b). In this section, we will first introduce the MAP measurements from the PACE mission and then review key components of the retrieval algorithm.

### 2.1    HARP MAP measurement

PACE will carry three instruments that are expected to advance our characterization of the atmosphere, ocean and land states (Werdell et al., 2019; Remer et al., 2019a, b; Frouin et al., 2019). The main instrument on PACE is a hyperspectral scanning

radiometer named the Ocean Color Instrument (OCI). There are two MAPs on PACE. The first is the Spectro-Polarimeter for Planetary EXploration one (SPEXone), contributed by a consortium of organizations in the Netherlands including SRON (Netherlands Institute for Space Research) and Airbbus Defense and Space Netherlands, which will perform multi-angle measurements at 5 along-track viewing angles of $0°$, $\pm20°$ and $\pm58°$, with a narrow cross-track nadir surface swath of 100 km, and a continuous spectral range spanning 385-770 nm at resolutions of 2-3 nm for intensity and 10-40 nm for polarization (van

Amerongen et al., 2019; Rietjens et al., 2019; Hasekamp et al., 2019a). The second is the Hyper-Angular Rainbow Polarimeter (HARP2), contributed by UMBC (University of Maryland, Baltimore County), a wide field-of-view imager that measures the total and polarized radiances at 440, 550, 670, and 870 nm, with a nadir-view swath of 1,556 km (Martins et al., 2018). The 670 nm band will measure 60 viewing angles compared to the other bands' 10 angles. AirHARP is the airborne version of HARP2 and measures the same number of viewing angles at 670 nm, but 20 viewing angles at the other three bands. Note that,

for the HARP instruments, the view angles observed by different spectral bands are close but not identical.

The total measured reflectance ($\rho_t(\lambda)$) and degree of linear polarization (DoLP; $P_t(\lambda)$) are taken as input to the FastMAPOL retrieval algorithm, defined as

$$\rho_t = \frac{\pi L_t}{\mu_0 F_0}, \tag{1}$$

$$P_t = \frac{\sqrt{Q_t^2 + U_t^2}}{L_t}, \tag{2}$$

where $L_t$, $Q_t$ and $U_t$ are the first three Stokes parameters, $F_0$ is the extraterrestrial solar irradiance, and $\mu_0$ is the cosine of the solar zenith angle. We adopt instrument calibration uncertainties of 3% in reflectance for both AirHARP and HARP2, 0.01 in DoLP for AirHARP, and 0.005 in DoLP for HARP2 (McBride et al., 2019; Puthukkudy et al., 2020; Gao et al., 2021a, b).

### 2.2    Neural network radiative transfer forward model

Vector radiative transfer models (VRTMs) are used to simulate the reflectance and polarization over a coupled atmosphere

and ocean system (Zhai et al., 2009, 2010). However, it is computationally time consuming to call a VRTM within a retrieval scheme, and the large number of retrieval parameters mean that creating a lookup table of results in reasonable size, as is common for lower-dimensionality retrievals, is prohibitive. Therefore, to achieve high speed and accuracy for retrievals, Gao



et al. (2021a) trained several feed forward neural network (NN) models with synthetic data generated by the VRTM developed by Zhai et al. (2009, 2010, 2022). NNs for reflectance ($\rho_t$) and DoLP ($P_t$) are trained individually, both with an input layer

with 15 parameters, followed by three hidden layers with 1024, 256 and 128 nodes, and a final output layer with 4 nodes to represent the four HARP bands. Details of the forward model and the NN training process are provided by Gao et al. (2021a).

The atmospheric model for the airborne measurements consists of a combination of aerosols and air molecules from surface to 2 km, an aerosol-free molecular layer (i.e. Rayleigh scattering) above that, and (for the airborne AirHARP instrument) an additional aerosol-free layer above the aircraft altitude. A total of 15 geophysical parameters, shown in Table 1, are used as

inputs to the forward model. The solar and viewing geometries are represented by the solar and viewing zenith angles ($\theta_0$ and $\theta_v$) and a relative azimuth angle ($\phi_v$). The aerosol complex refractive index for both fine and coarse modes is assumed to be spectrally flat, represented by 4 parameters, including both real ($m_{r,f}$ and $m_{r,c}$) and imaginary ($m_{i,f}$ and $m_{i,c}$) parts. The aerosol size distribution is assumed as a combination of five lognormally-distributed aerosol sub-modes, each with prescribed mean radii and variances; the five volume densities ($V_i$) are free parameters (Dubovik et al., 2006; Xu et al., 2016). The

combined aerosol fine mode consists of the three smaller sub-modes, and the coarse mode the two larger sub-modes. Therefore, the fine mode volume fraction (fvf) is defined as:

$$\text{fvf} = \frac{\sum_{i=1}^{3} V_i}{\sum_{i=1}^{5} V_i} \tag{3}$$

Ozone absorption is quantified by the ozone column density ($n_{O3}$); absorption by other gaseous species is minimal in HARP's spectral bands and is therefore neglected. Ocean surface roughness is represented by the isotropic Cox and Munk

model (1954) parameterized by wind speed (m s$^{-1}$). Strong sunglint is excluded here by removing view angles within $40°$ of the specular reflection direction due to the challenges to represent the sunglint signals from ACEPOL field campaign using the isotropic Cox and Munk model (Gao et al 2020, Gao et al 2021a). We only consider open ocean waters modeled as a uniform layer with bio-optical properties parameterized as a function of chlorophyll-a concentration (Chla) (Gao et al., 2019). Complex bio-optical properties for coastal waters require additional parameters in the bio-optical model (Gao et al., 2018), which require

additional NN trainings that will be pursued in a future study.

NN uncertainties $\sigma_{NN}$ are $< 1\%$ for reflectance and $< 0.003$ for DoLP for all HARP bands, which are much smaller than the measurement uncertainties (Sec. 2.1). To achieve high NN accuracy, numerical uncertainty on the radiative transfer simulations used to train the NN has an uncertainty $\sigma_{RT}$ much smaller than $\sigma_{NN}$ (Gao et al., 2021a). The forward calculation of aerosol size and refractive index to aerosol optical depth (AOD) and single scattering albedo (SSA) is also performed using NNs based on

simulations using the numerical code based on the Lorenz-Mie theory (Mishchenko et al., 2002). In addition, the spectral ocean color remote sensing reflectance ($R_{rs}(\lambda)$) is derived based on the retrieved aerosol properties through atmospheric correction (Mobley et al., 2016; Mobley, 2022) and are also implemented here with NNs. The performance of all NNs has been quantified and reported by Gao et al. (2021a).





**Table 1.** Parameters used to train the FastMAPOL forward model as described in Sec2.2. The minimum (min) and maximum (max) values of each parameter are also shown. The a priori uncertainties ($\sigma_a$) are estimated as the difference between the max and min values for the study in Sec. 3, except the four parameters as indicated which are assumed as known input.

| Parameters | Unit | Min | Max | $\sigma_a$ |
|---|---|---|---|---|
| $\theta_0$ | ° | 0 | 70 | [input] |
| $\theta_v$ | ° | 0 | 60 | [input] |
| $\phi_v$ | ° | 0 | 180 | [input] |
| $n_{O3}$ | DU | 150 | 450 | [input] |
| $V_1$ | $\mu m^3 \mu m^{-2}$ | 0 | 0.11 | 0.11 |
| $V_2$ | $\mu m^3 \mu m^{-2}$ | 0 | 0.05 | 0.05 |
| $V_3$ | $\mu m^3 \mu m^{-2}$ | 0 | 0.05 | 0.05 |
| $V_4$ | $\mu m^3 \mu m^{-2}$ | 0 | 0.19 | 0.19 |
| $V_5$ | $\mu m^3 \mu m^{-2}$ | 0 | 0.58 | 0.58 |
| $m_{r,f}$ | (None) | 1.3 | 1.65 | 0.35 |
| $m_{r,c}$ | (None) | 1.3 | 1.65 | 0.35 |
| $m_{i,f}$ | (None) | 0 | 0.03 | 0.03 |
| $m_{i,c}$ | (None) | 0. | 0.03 | 0.03 |
| $w$ | $ms^{-1}$ | 0.5 | 10 | 9.5 |
| Chla | $mg \cdot m^{-3}$ | 0.01 | 10 | 10 |

## 2.3 Cost function and input uncertainty model

The optimal values of retrieval parameters are obtained using a maximum likelihood approach by minimizing the difference between the measurements and the forward model fit represented by a cost function (Rodgers, 2000):

$$\chi^2 \quad = \quad \frac{1}{N}[\mathbf{F}(\mathbf{x}) - \mathbf{m}]^T \mathbf{S}_\epsilon^{-1}[\mathbf{F}(\mathbf{x}) - \mathbf{m}], \tag{4}$$

where $\mathbf{m}$ is a vector including measurements from all angles and bands (both total reflectance and DoLP; Eqs. 1 and 2) and $\mathbf{F}(\mathbf{x})$ is the forward-modeled observations described in the previous section. The state vector $\mathbf{x}$ includes the 11 parameters

retrieved as summarized in Table 1. $N$ is the total number of measurements. The input uncertainty model is characterized by the error covariance matrix $\mathbf{S}_\epsilon$ representing the combined measurement and forward model uncertainty. In this work, we assume uncorrelated uncertainty and therefore $\mathbf{S}_\epsilon$ is a diagonal matrix (Part 2 of this work will assess the effect of true and assumed correlated uncertainties on the retrieval and propagated uncertainty estimates). The diagonal elements ($\sigma_\epsilon$) include contributions from instrumental $\sigma_{ins}$, neural network $\sigma_{NN}$, and VRTM $\sigma_{RT}$ assuming no correlations between these uncertainty sources:

$$\sigma_\epsilon^2 \quad = \quad \sigma_{ins}^2 + \sigma_{NN}^2 + \sigma_{RT}^2. \tag{5}$$

  The subspace trust-region interior reflective (STIR) algorithm is used to conduct non-linear least-square minimization of the cost function (Branch et al., 1999; Virtanen et al., 2020). STIR is based on the Levenberg-Marquardt algorithm combined with



an interior method and reflective boundary method to ensure the retrieval parameters are well searched within their ranges as specified in Table 1 (Branch et al., 1999).

## 3 Uncertainty quantification for MAP retrievals

### 3.1 Pixel-wise retrieval uncertainty quantification

The propagated (theoretical) pixel-wise uncertainty quantification is based upon a Bayesian approach which assumes Gaussian distributions of input uncertainty (including measurements, forward model and a priori) and output (retrieval) uncertainty (Rodgers, 2000). These represent the one standard deviation ($1\sigma$) uncertainty on the retrieved state, and are determined by mapping the measurement and forward model uncertainties into retrieval parameter space,

$$\mathbf{S^{-1}} = \mathbf{K^T S_\epsilon^{-1} K} + \mathbf{S_a^{-1}}, \tag{6}$$

where $\mathbf{S}$ is the retrieval uncertainty covariance matrix, $\mathbf{S_\epsilon}$ is the error covariance matrix as in Eq. 4 which includes contributions from measurement and forward model as shown in Eq. 5, $\mathbf{K}$ is the Jacobian matrix, and $\mathbf{S_a}$ is the *a priori* uncertainty covariance matrix. FastMAPOL does not use explicit a prior information on the cost function. However, each retrieval parameter has a range of acceptable values (Table 1) which are imposed by the STIR optimization algorithm, therefore these parameter ranges work as implicit prior constraints. To capture the impact of these constraints, we assume Sa is diagonal and take the permitted range of each state parameter as an assumed a prior uncertainty as listed in Table 1. This is an approximation to the Rodgers (2000) formalism and serves to stop the retrieval uncertainty exceeding the physically-plausible range (though in most cases it has little numerical effect). The Jacobian matrix, K, expresses the sensitivity of the forward model to changes in the retrieval parameters, which is defined as

$$K_{ij}(\mathbf{x}) = \frac{\partial \mathbf{F}_i(\mathbf{x})}{\partial x_j}, \tag{7}$$

where indices $i$ and $j$ represent the different measurements and the retrieved parameters, respectively. The finite difference method is often used to compute the Jacobian matrix, but it is time-consuming due to the many retrieval parameters used to calculate the derivatives. In our previous work (Gao et al., 2021b), we implemented an analytical approach based on neural networks, which is extended here with significant speed improvement as discussed in Sec. 3.3.

The $1\sigma$ uncertainties on each retrieved parameter are simply the square roots of the diagonal elements of $\mathbf{S}$. For quantities $x_a$ that are not directly contained in x but can be calculated from it, such as AOD or SSA, their uncertainty ($\Delta_a$) can be expressed as:

$$\Delta_a = \sqrt{\sum_i \sum_j \mathbf{S}_{i,j} \frac{\partial a}{\partial x_i} \frac{\partial a}{\partial x_j}}. \tag{8}$$

The additional derivatives of $x_a$ with respect to state parameters necessary to compute $\Delta_a$ are also calculated using the efficient analytical expressions based on NN models as discussed in Sec. 3.3.




Other Bayesian inference methods exist that are capable of deriving retrieval uncertainties without explicitly computing the Jacobian matrix or requiring that uncertainties be Gaussian. For example, Knobelspiesse et al. (2021) applied the Generalized Nonlinear Retrieval Analysis (GENRA, Vukicevic et al. (2010)) method on simulated MISR data to access the retrieval uncer­tainties of multiple retrieval parameters. However, such methods often require a large number of computationally-expensive forward model calculations, and are less practical for high dimensional problems such as this. Thus, the more computationally efficient Jacobian-based approach is the main focus of this work.

### 3.2 Retrieval uncertainty performance evaluation

Verifying theoretical uncertainty estimates is necessary because real retrieval performance depends on other factors. A key factor is how well the inversions converge to the global minimum of the cost function instead of a false convergence to a local minimum. This is not captured by Eq. 6. Several factors can lead to false convergence to local minima, e.g.:

- accuracy of the forward model and Jacobian matrix,

- tolerance for iterative optimization, which may impact how early the iterative parameter updates stop,

- retrievals may get stuck at parameter boundaries, if not adequately treated in the inversion algorithm,

- the input uncertainty model may be insufficient, leading to inappropriate weights of different measurements in the cost function,

- false convergence from non-monotonic cost functions due to insufficient information in the measurements.

To evaluate the performance of the uncertainty quantification using error propagation, we can compare theoretical uncertainty with the uncertainties calculated by comparing the final retrieval results with reference truth values. Two useful metrics, the mean absolute error (MAE), and the root mean square error (RMSE) between the truth ($T_i$) and retrievals ($R_i$), are defined as

$$\text{RMSE} \quad = \quad \sqrt{\frac{1}{M}\sum_{i=1}^{M}(R_i - T_i)^2}, \tag{9}$$

$$\text{MAE} \quad = \quad \frac{1}{M}\sum_{i=1}^{M}|R_i - T_i|, \tag{10}$$

where $M$ is the total number of retrieval cases. Note that for a Gaussian distribution, RMSE and MAE are related as:

$$\text{RMSE} = \sqrt{\pi/2}\text{MAE}, \quad \text{for a Gaussian distribution} \tag{11}$$

MAE is more robust to outliers than RMSE, so comparing the two can be informative as to whether the overall error distribution is close to Gaussian. MAE has also been shown to be less dependent on the number of cases considered than RMSE (Willmott and Matsuura, 2005). Note that, over a large ensemble of cases, the overall error distribution is not necessarily expected to be Gaussian because it may be drawn from a large number of different atmospheric/oceanic states, each with a different magnitude of uncertainty.





Chlorophyll a concentration (Chla) varies across several orders of magnitude, and plays an important role to determine $R_{rs}$ and their the uncertainties (McKinna et al., 2019). As recommended by Seegers et al. (2018), we use a log-transformed metric:

$$\text{MAE(log)} = 10^Y \text{ where } Y = \frac{1}{M} \sum_{i=1}^{M} |\log_{10}(R_i) - \log_{10}(T_i)|. \tag{12}$$

MAE(log) indicates the averaged ratio between the retrieval and truth values in such a way that a value of 1.2 indicates that the retrievals exceed truth by 20%. To compare with the theoretical uncertainty for Chla requires that its retrieval uncertainty must

be transformed to a log 10 scale as follows:

$$\Delta\log_{10}(\text{Chla}) = \frac{\Delta\text{Chla}}{\text{Chla} \cdot \ln 10} \tag{13}$$

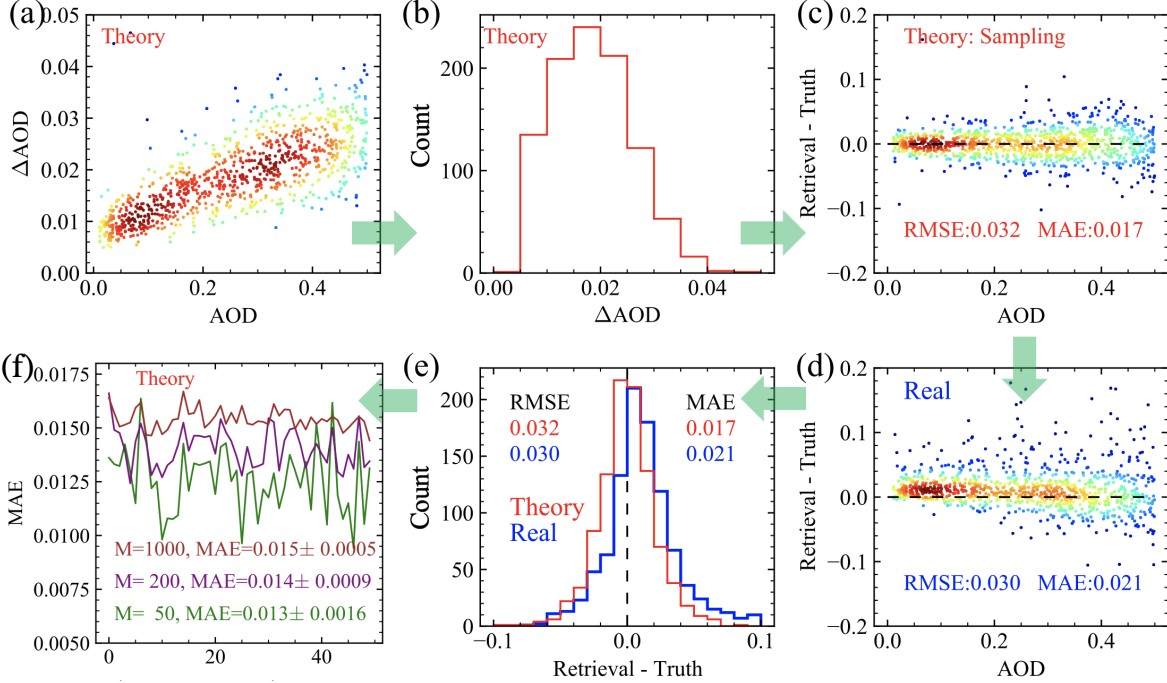

**Figure 1.** Demonstration of the procedures to compare theoretical and real uncertainties. (a) theoretical uncertainties of AOD retrievals over 1000 synthetic HARP2 measurements; (b) histogram of (a); (c) the retrieval error sampled from (a); (d) the retrieval error derived from the difference between the real retrieval results and truth data; (e) the histogram for the retrieval errors in both (c) and (d); (f) The MAE for 50 set of random theoretical errors considering a total number of cases of 50, 200, and 1000.

Direct comparison of theoretical uncertainties and real errors is difficult because the former is a measure of the estimated dispersion of the retrieval in terms of a distribution of $1\sigma$ uncertainties, and the latter is a distribution of retrieval errors indicating the difference between real retrieval results and the truth reference data that relate specifically to observational

conditions available at the time of collection. To effectively compare the theoretical uncertainties and real errors, we propose a



sampling-based method, Monte Carlo Error Propagation (MCEP), which samples random retrieval errors from the theoretical uncertainties and therefore enables comparisons on the same retrieval error domains. This method is demonstrated in Fig. 1 using 1000 synthetic retrievals of AOD at 550 nm from HARP2 data. The synthetic datasets are generated with random draws from a uniform distribution of AOD values from 0.01 to 0.5. The selection of a uniform AOD distribution is to ensure the same

number of cases are considered in each sub-interval for later statistical discussion. Detailed information on the synthetic data is provided in the next section. This choice of synthetic data is to explore the dependency of retrieval uncertainties with respect to AOD. To represent the overall retrieval performance of actual PACE data, synthetic or real HARP2 data with realistic statistical distributions will be studied in the future.

The goal is to generate a statistical distribution of the retrieval error (defined as the difference between retrieval and truth)
for both theoretical and real uncertainties and develop proper metrics for comparison based on the distribution. Steps involved in MCEP are listed below using the example in Fig. 1:

1. Conduct retrievals and compute theoretical retrieval uncertainties according to the error propagation method discussed in Sec. 3.1. Here AOD is derived from the directly retrieved refractive indices and volume densities shown in Table 1, and $\Delta$AOD is thereby calculated from Eq. 8 for each individual retrieval. Fig. 1(a) shows the theoretical AOD uncertainties
evaluated for 1000 cases with its histogram shown in Fig. 1(b).

2. Generate a distribution of random theoretical errors. This is done by taking the theoretical uncertainty for each retrieval and generating a random number from a Gaussian distribution with a zero mean and a standard deviation equal to the theoretical uncertainty (i.e. individual points from Fig. 1(a)). This random number will be the theoretical retrieval error for the corresponding theoretical retrieval uncertainty. These sampled random errors are shown in Fig. 1(c).

3. The real retrieval errors, shown in Fig. 1(d), are calculated as the difference between the retrieval results and truth data. Fig. 1 (c) and (d) showed similar dependency on the AOD.

4. The histograms for the error data in (c) and (d) are compared in (e), which shows statistical distributions directly comparable. These distributions can be analyzed using metrics such as RMSE and MAE in Eqs. 10 and 11.

5. Evaluate the variations of the uncertainty metrics derived from step 4 : 1) Generate multiple sets of random theoretical
errors following step 2; 2) Compute the metrics for each set of errors ; 3) Compute $1\sigma$ uncertainties of the metrics. This uncertainty depends on the number of cases used within each set and, therefore, can also be used to approximate the uncertainty of the metrics evaluated from real errors due to the same number of cases (M) used in Eq. 9 and 10. The MAE results for M equals to 50, 200, 1000 over 50 sets of theoretical random errors are shown in (f).

The MCEP method enables direct comparison of error distributions between theoretical uncertainties and real retrievals,
which therefore provide additional flexibility in analyzing their statistics. For the example in Fig. 1(e), the peak of real retrieval errors is ~0.01, suggesting that the retrievals tend to overestimate total AOD. The sampling method used in step (2) of MCEP does not assume any particular statistical distribution of the AOD values and their theoretical uncertainties. The random sampled error distribution, similar to the real errors, is more peaked than a Gaussian distribution, with ratios between RMSE and





MAE of $0.032/0.017 = 1.88$ and $0.030/0.021 = 1.43$ for the real and theoretical errors, respectively. The larger ratios (com-
pared to 1.25 for a Gaussian, Eq. 11) confirm that both distributions have a narrower peak and longer tails (therefore larger
RMSE values) than a Gaussian distribution. To evaluate the retrieval uncertainties quantitatively and reduce the influence of
outliers, in later studies, we focus on MAE evaluated from the random errors as shown in Fig 1(e). Since the MCEP method is
directly based on the statistical distribution, metrics other than MAE and RMSE can also be derived. For example, the method
proposed by Sayer et al. (2020), which computes the 68th percentile from absolute normalized error distributions, can be ap-
plied on the random error samples in the MCEP method as a metric to evaluate $1\sigma$ uncertainties for both real and theoretical
errors.

Furthermore, following step 5 in MCEP, we can analyze the uncertainties of MAE with respect to a set of random errors.
MAE values for 50 sets of random theoretical errors are computed as shown in Fig. 1(f). The relative standard deviation of these
MAE values is about 3% when all 1000 cases are used. The relative uncertainties increase to 7% and 12% when the number
of cases are reduced to 200, and 50. Therefore, for discussion in the next section with a smaller number of cases considered, it
is useful to understand how much the MAE varies. A similar approach can be applied to comparisons with high-quality in-situ
measurements. The same challenge is that the metrics such as RMSE and MAE may suffer from larger statistical variations if
only a smaller number of retrieval cases are available.

### 3.3 Speed improvement using automatic differentiation

Gao et al. (2021b) found that using the automatic differentiation method to compute Jacobians resulted in a factor of 5 to 10
speedup in retrievals compared to numerical calculations using finite-difference. This method is applied here as well for the
computation of retrieval uncertainties. Additional derivatives must be calculated for parameters derived from directly-retrieved
state quantities as shown in Eq. 8. Such parameters in this study include aerosol properties such as AOD, SSA, aerosol effective
radius, and $R_{rs}$. Derivatives of aerosol properties are often simple as they can be either computed from an analytical function
(e.g. effective size) or based on single scattering calculations (e.g. AOD, SSA), such as using Lorenz-Mie theory (Grainger
et al., 2004; Spurr et al., 2012) or the T-Matrix method (Xu and Davis, 2011; Spurr et al., 2012; Sun et al., 2021). Uncertainties
for $R_{rs}$ are more challenging to quantify as they require additional radiative transfer simulations to conduct atmospheric and
bidirectional reflectance distribution function (BRDF) corrections. Following Mobley et al. (2016), $R_{rs}$ is defined as:

$$R_{rs} = \left[\frac{\rho_t - \rho_{t,\text{atm+sfc}}^f}{\pi}\right] \times \left[\frac{C_{\text{BRDF}}}{T_d t_u}\right], \tag{14}$$

where $\rho_t$ is the reflectance measured by the sensor as defined in Eq. 1, $\rho_{t,atm+sfc}^f$ is the reflectance with contributions only
from the atmosphere and ocean surface. $C_{BRDF}$ is a BRDF correction that adjusts the water leaving signal from an arbitrary
viewing and solar geometry to the sun at zenith and nadir viewing direction. $T_d$ and $t_u$ are direct and diffuse transmittance.

To achieve a fast uncertainty evaluation speed, equivalent to the fast retrieval speed, we use automatic differentiation to
calculate analytical Jacobians and other derivatives for AOD, SSA, $\rho_{t,atm+sfc}^f$, and the combined factor of $C_{BRDF}/[T_d t_u]$
based on the NNs developed by Gao et al. (2021a). The mathematical formulation for automatic differentiation summarized in
Gao et al. (2021b) can be generalized for all the feed-forward neural networks used in our study. Specifically, the derivatives





of Rrs with respect to a retrieval parameter $x_i$ are

$$\frac{\partial R_{rs}}{\partial x_i} = \frac{(\rho_t - N_1)}{\pi}\frac{\partial N_2}{\partial x_i} - \frac{\partial N_1}{\partial x_i}\frac{N_2}{\pi}, \tag{15}$$

where $N_1$ and $N_2$ represent the NNs for $\rho_{t,atm+sfc}^f$ and $C_{BRDF}/[T_d t_u]$. The uncertainty of Rrs is calculated by combining Eq.
15 with Eq. 8. Note that the retrieval uncertainties in $R_{rs}$ discussed in this study only include the contribution from atmospheric correction and BRDF correction as shown in Eq. 15, which do not include uncertainties in $\rho_t$. These results can demonstrate the accuracy when HARP retrieved aerosol properties are applied to instruments with higher accuracy in $\rho_t$ such as OCI to assist their atmospheric correction (Gao et al., 2020; Hannadige et al., 2021).

## 4   Retrieval uncertainties from synthetic AirHARP and HARP2 measurements

To evaluate the retrieval capability of the FastMAPOL algorithm on the HARP instruments, we conducted studies on synthetic AirHARP and HARP2 data, and then derived the pixel-wise retrieval uncertainties. The theoretical uncertainties are then compared with real uncertainties and their difference are quantified using the MCEP methodology discussed in Sec. 3. Note that the real uncertainties are derived from the retrieval results based on synthetic data which include impacts from local minima in the cost functions as summarized in Sec. 3.2, however, these synthetic data studies do not address the potential impacts of modeling errors in the forward model. To evaluate the assumption in the forward model, comparison with in-situ measurements is required in future studies.

### 4.1   Synthetic data

We performed radiative transfer simulations to generate 1,000 synthetic sets of measurement using the coupled atmosphere-ocean VRTM (Zhai et al., 2009, 2010, 2022) discussed in Sec. 2. A fixed solar zenith angle of $50°$ is used to approximate the solar zenith angle from the AirHARP measurements in the ACEPOL field campaign discussed in the next section. The other input parameters in Table 1 are sampled uniformly within their ranges, except aerosol volume densities and Chla. Aerosol volume densities are determined by AOD at 550 nm, which is sampled uniformly over the range [0.01, 0.5], and fine mode volume fraction, sampled uniformly within [0, 1]. Chla is randomly sampled with a log-uniform distribution. Although ozone density is randomly sampled to generate synthetic data, it is assumed as known input to the retrieval algorithm.

Realistic HARP-like viewing geometries are constructed as discussed in Gao et al. (2021b) which represents a simplified PACE orbit geometry with some examples in Fig. 2 (a). The number of viewing angles at each band is based on AirHARP and HARP2 characteristics (Sec. 2.1).





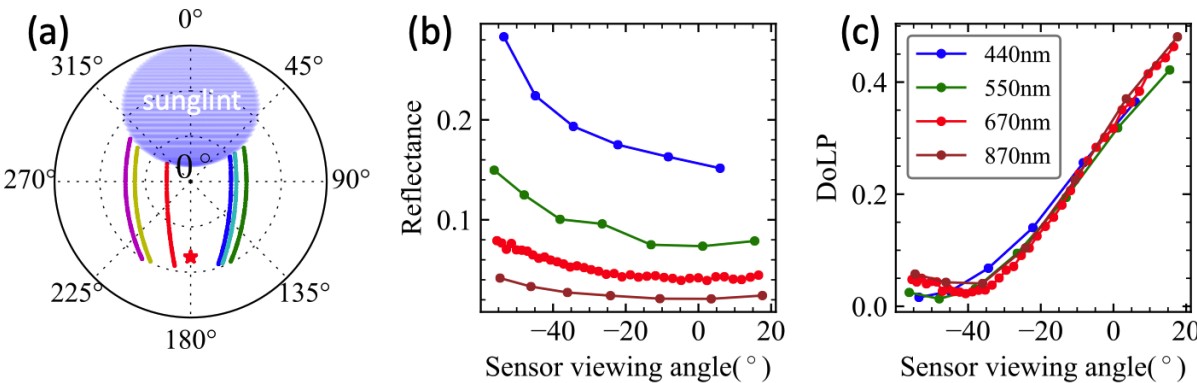

**Figure 2.** (a) Random viewing geometry with different examples indicated by different colors in the polar plot of zenith angles (radial direction) and azimuth angles. The red star symbol indicates the anti-solar direction at a zenith angle of $50°$ and azimuth angle of $180°$. The blue oval shape indicates the sunglint region removed in this study. (b) and (c) Example synthetic HARP2 data with added random noise for reflectance and DoLP. The sensor viewing angle indicates the viewing zenith angle in the along-track direction with positive angle indicating forward-looking directions, and a negative sign indicating the angle viewing backward (within the azimuthal angle between $90°$ and $270°$)

Random noise is added to the 1000 sets of synthetic AirHARP and HARP2 measurements and then the FastMAPOL retrieval algorithm is applied to them. Note that the synthetic data is computed directly using the vector radiative transfer model, but the NN forward model is used in the retrieval algorithm. The retrieval cost function values ($\chi^2$) at convergence (Eq. 4) are shown for both sensors in Fig. 3; the mean $\chi^2$ values for both cases are approximately 1.0, but with the most probable $\chi^2$ values being 0.8 for HARP2 and 0.9 for AirHARP, which suggests slight overfitting of the data in general.

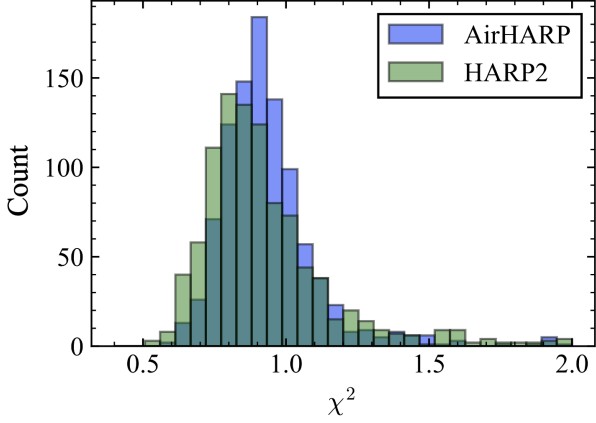

**Figure 3.** The histogram of the cost function values for the synthetic retrievals.



## 4.2 Pixel-wise retrieval uncertainties quantification

We apply the method discussed in Sec. 3 to compare theoretical and real uncertainties. An example of spectral AOD and $R_{rs}$

for one retrieval is shown in Fig. 4 with the retrieval uncertainties as a function of wavelength. Here, total AOD uncertainty is dominated by the fine mode but relative uncertainty on the coarse mode is larger. The absolute $R_{rs}$ uncertainties at 440 nm and 550 nm are larger than at 670 nm and 870 nm, as are the errors (i.e. differences between the retrieval and truth data). However, the $R_{rs}$ percentage errors generally increase with the wavelength due to the decrease of the $R_{rs}$ magnitudes.

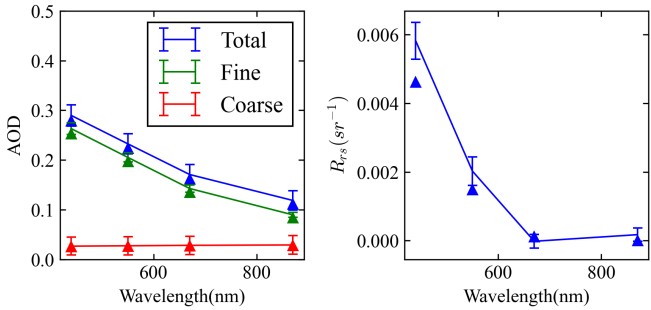

**Figure 4.** Example of AOD (solid line) and Rrs spectra retrieved from one case of synthetic HARP2 measurements as shown in Fig 1(b) and (c) with retrieval uncertainties indicated by the error bar. The triangles indicate truth values. Chla for this case is 0.1 $mgm^{-3}$.

For more general atmosphere and ocean conditions, Fig. 5 shows dependence of the retrieval uncertainties on AOD at 550

nm for retrieved and derived parameters from synthetic HARP2 measurements. In general, increasing AOD is associated with increasing AOD uncertainty. The uncertainty of ocean parameters also increases with AOD, which is expected because the atmosphere is an obstruction to the oceanic signal. Increasing AOD does, however, decrease the uncertainty of retrieved and derived aerosol properties. These changes are not always a linear function of AOD. The larger spread of coarse mode properties (particularly SSA) than fine mode results indicate less sensitivity to coarse mode aerosol property retrievals.



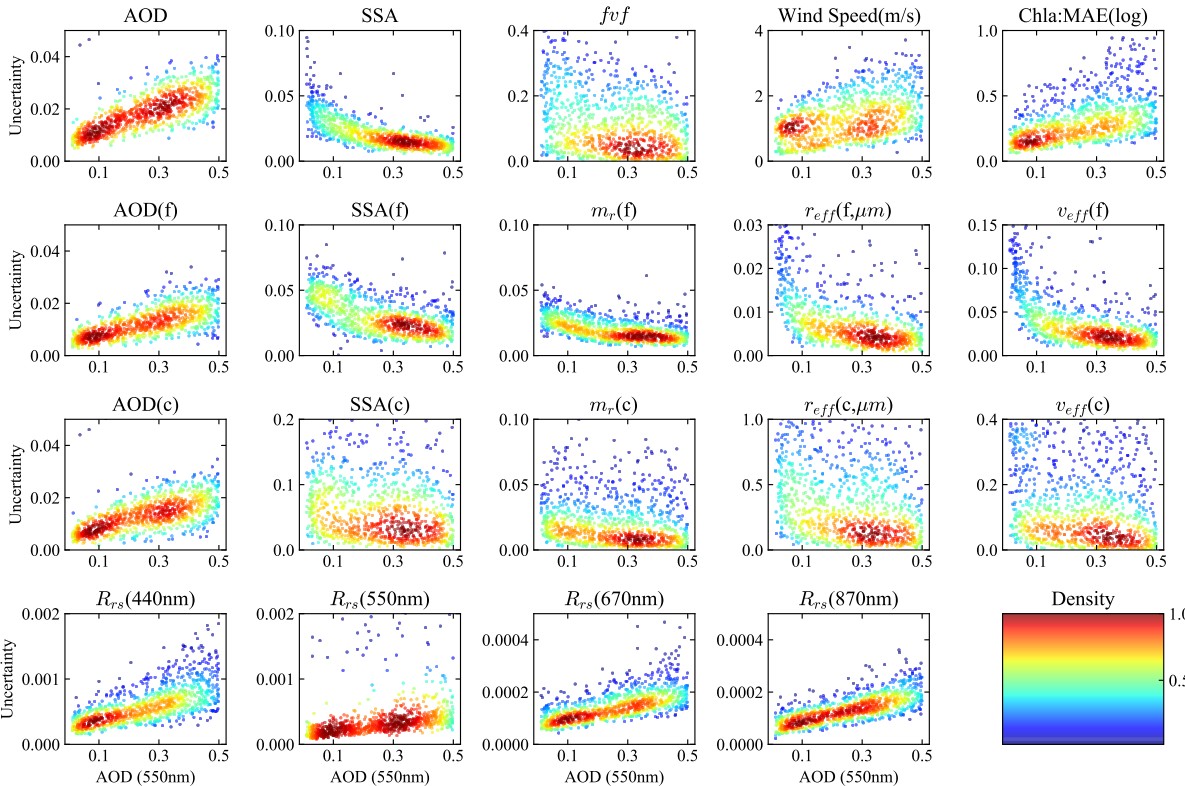

**Figure 5.** Theoretical retrieval uncertainties estimated from error propagation plotted against the AOD at 550 nm (horizontal axis) for AOD, SSA, fine mode volume fraction(fvf), refractive index ($m_r$), effective radius ($r_{\text{eff}}$) and variance ($v_{\text{eff}}$), wind speed, Chla in log 10 scale as well as remote sensing reflectance ($R_{rs}$). Synthetic HARP2 measurements are used in these retrievals. Colors indicate the relative density of the dots in the plot.





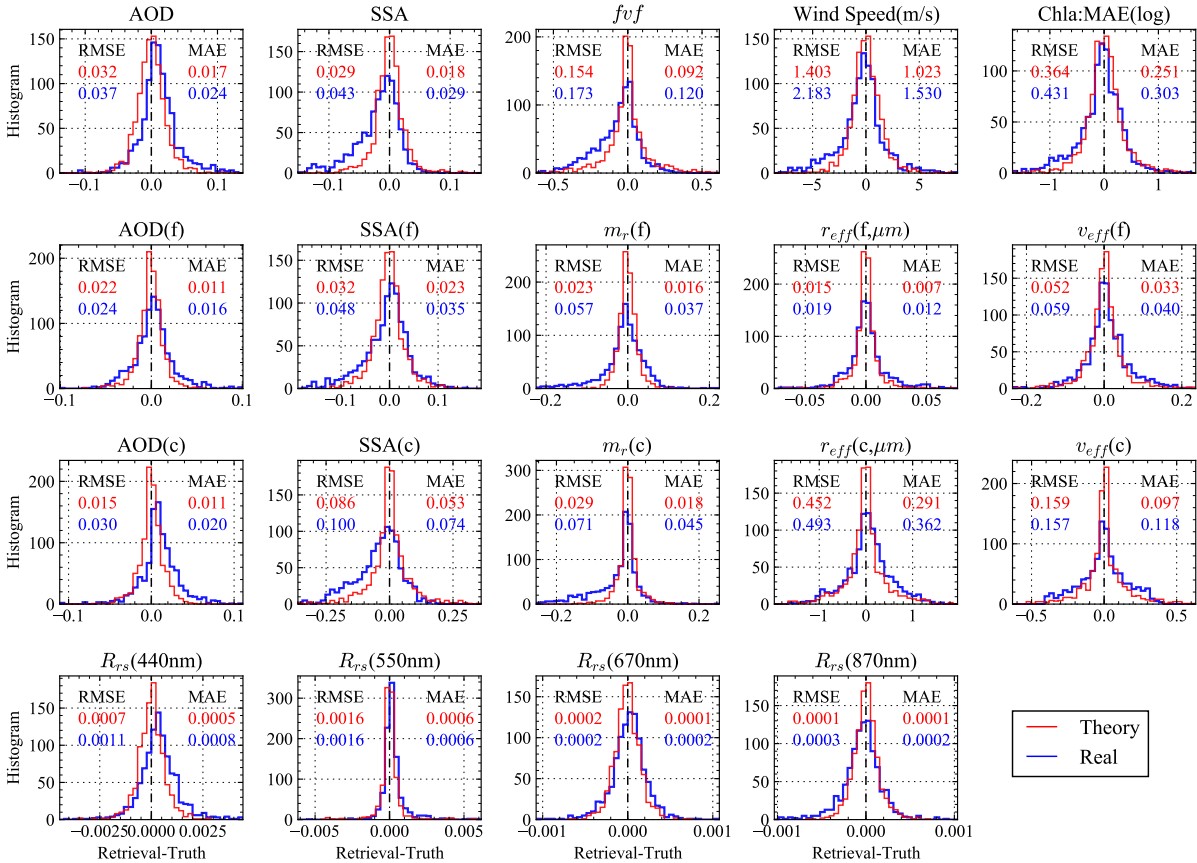

**Figure 6.** Histograms of the theoretical and real retrieval errors evaluated using the MCEP method in Sec. 3.2 for the same cases as in Fig. 5

Following the methodology proposed in Sec. 3.2, the statistical distributions of the retrieval errors are shown in Fig. 6 derived from the theoretical retrieval uncertainties in Fig. 5. Most histograms show a distribution with a well-centered peak and similar width and shape between the theoretical and real uncertainties. The mean value indicates the bias of the distribution. The AOD error distribution has a slightly longer tail in the positive side, resulting in a mean difference of 0.011 for both total and coarse mode AOD; the mean value difference for fine mode AOD is negligible (0.001) (also discussed in Fig. 4). These results suggest

that the source of the bias in total AOD is due to the impacts from coarse mode retrievals. Similar to AOD, most distributions in Fig. 6 are narrower than a Gaussian distribution with longer tails, and the ratios of RMSE and MAE from both theoretical and real uncertainty results are mostly between 1.3 and 2. The histogram of the wind speed error over the ensemble seems to be closer to Gaussian. SSA has a relatively larger negative tail mean values of -0.02, -0.01, and -0.04 for total, coarse mode, and fine mode SSA. Refractive index differences also show a larger negative tail indicating a trend of slightly underestimating the

refractive index, which leads to a mean value of -0.01 and -0.03 for the fine and coarse mode real refractive indices. However, the most probable errors for refractive index are well centered around zeros.





### 4.3 Evaluating the performance of pixel-wise retrieval uncertainty

To quantify theoretical and real uncertainties, Fig. 7 shows MAE for AirHARP and HARP2 averaged as a function of AOD at

550 nm, based on the error distributions shown in Fig. 6. The uncertainties of the total, fine and coarse mode AOD increase as

385  AOD increases, though the ratio of AOD uncertainty to AOD shows a decreasing trend. As in Fig. 5, uncertainties of aerosol

microphysical properties (SSA, refractive index, effective radius and variance) decrease as AOD increases, which is consistent

with Gao et al. (2021a). The uncertainty for Chla is represented in terms of MAE(log) as defined in Eq. 12 with a value between

1 and 3 which also depends upon the magnitude of Chla as discussed in Gao et al. (2021a). The uncertainty of Rrs increases

almost linearly with AOD. At 440 nm, the uncertainty increases from 0.0004 to 0.0012, while for 550 nm, the uncertainty

increases from 0.0002 to 0.0007. Note that the accuracy of the atmospheric correction used to derive Rrs also depends upon

the number of viewing angles used for aerosol retrievals (Gao et al., 2021b).

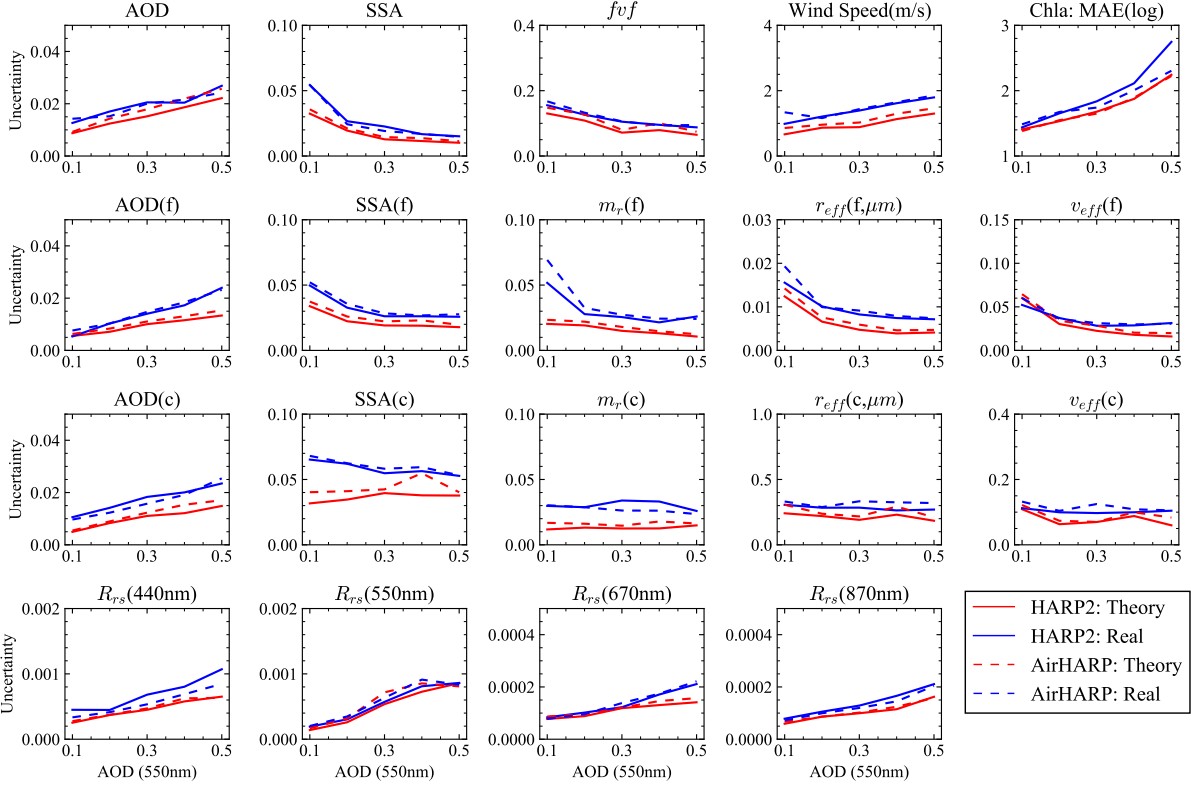

**Figure 7.** The retrieval uncertainties represented by MAE averaged within several range of AOD at 550nm, including [0.01, 0.1], [0.1, 0.2], [0.2, 0.3], [0.3, 0.4], [0.4, 0.5]. The horizontal axis indicate the maximum AOD used in the corresponding AOD range. Results for both HARP2 and AirHARP are shown. Chla in terms of MAE(log) as defined in Eq.12 is used.

The retrieval uncertainties for synthetic HARP2 and AirHARP datasets are close to each other for most retrieval cases as shown in Fig. 7. Gao et al. (2021b) demonstrated that HARP2 has a smaller retrieval uncertainty than AirHARP when the





same number of viewing angles are used due to HARP2's smaller DOLP calibration uncertainty. However, this is partially

compensated by AirHARP's higher number of view angles, resulting in similar retrieval uncertainties for the two sensors in

Fig. 7. Note that the uncertainty correlation between angles may also impact the retrieval performance which are not included

in this study.

## 4.4   Averaged retrieval uncertainty

To understand the accuracy of the MAE as derived above for each AOD range (each with around 200 cases), we generated

multiple sets of random theoretical errors following step 5 in Sec. 3.2 and compared the averaged MAE with the MAE derived

from real errors as shown in Fig. 8. Most relationships are linear and close to the 1:1 line, indicating that the retrieval is skillful

at determining magnitudes as well as which retrievals are better-constrained than others). The exception is coarse mode aerosol

properties which tend to cluster together due to less dependency on the total AOD as shown in Fig. 7. The $1\sigma$ uncertainties of

the MAE for theoretical uncertainties are also shown in Fig. 8 as the horizontal error bar for both HARP2 and AirHARP. 10 sets

of random errors are found sufficiently to evaluate the uncertainties for MAE. We found that MAE varied within approximately

10% of its mean value in most cases, except for coarse mode properties, wind speed and Rrs at 550 nm which can reach up

to 15%. The same values are used to estimate the uncertainties of the real errors due to the impact of the number of cases.

Therefore, the MCEP method can assess the impact of the number of cases for comparison with in-situ measurement in future

studies, where satellite/ground match-up availability can vary dramatically depending on the location of in-situ site (Sayer

et al., 2020).



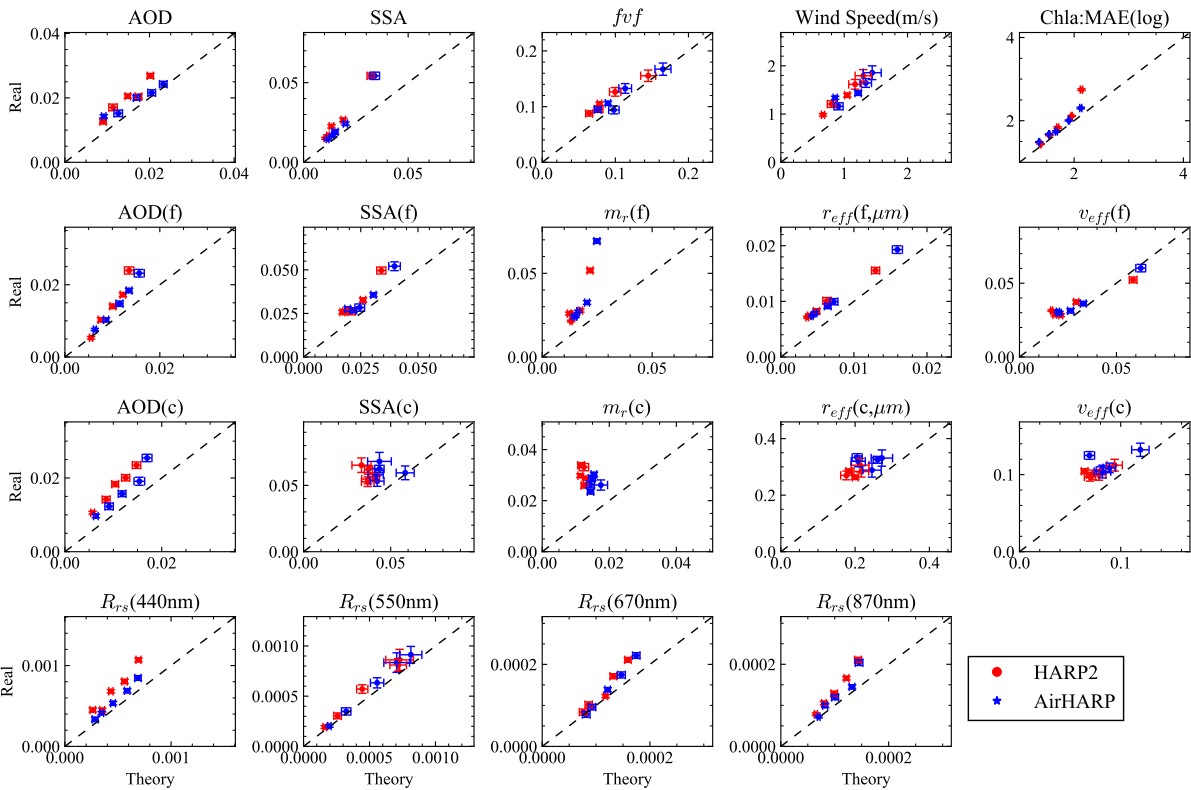

**Figure 8.** Comparing the averaged MAE derived from theoretical and real uncertainties for both HARP2 and AirHARP. The error bars indicate the $1\sigma$ uncertainties of the MAE based on the average of 10 sets of random theoretical errors as discussed in MCEP method in Sec. 3.2. The same error bar is used for the real uncertainties as an approximation

Ratios between the averaged MAEs for the real and theoretical uncertainties over five AOD intervals from 8 are shown in Fig. 9. The ratios are mostly in the range 1-1.5, indicating that the theoretical uncertainties work well to represent the real retrieval uncertainties in most cases but are generally slight underestimates. The largest ratios are for fine and coarse mode aerosol refractive indices, especially at small aerosol loading probably due to the lack of information and therefore more impact of local minima and initial values (Hasekamp and Landgraf, 2005). The large ratios of real and theoretical uncertainties also indicate where retrieval algorithms can be further improved, for instance, by including additional a priori constraints (Dubovik et al., 2021). A similar methodology can be applied to validate the retrieval performance of future space-borne sensors such as HARP2 measurements from PACE, with more realistic parameter distributions.



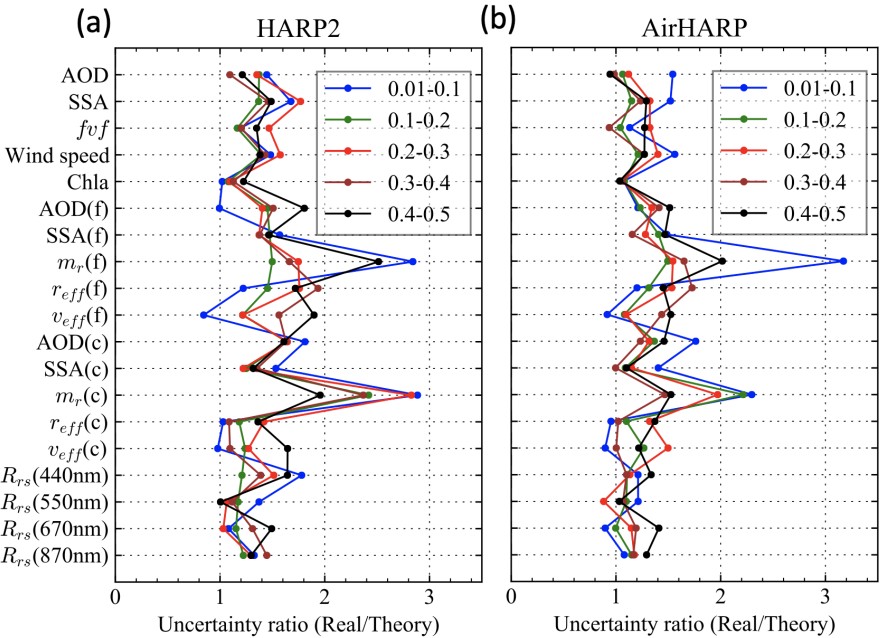

**Figure 9.** Ratio of real to theoretical retrieval MAE for the data shown in Fig. 8. Chla is in terms of MAE(log) as defined in Eq.12.

## 5  Retrieval uncertainties from AirHARP field measurements

The pixel-wise theoretical uncertainties achieve a reasonably good performance to represent real retrievals as discussed in the last two sections. Their performances on various retrieved geophysical properties are quantified by comparing with the real retrieval errors. Based on these results, in this section, we will use the theoretical uncertainties to analyze the retrieval results from AirHARP field measurements from the Aerosol Characterization from Polarimeter and Lidar (ACEPOL) field campaign conducted from October to November of 2017, where the NASA's ER-2 aircraft carried four MAPs: AirHARP, AirMSPI,

SPEX airborne, and RSP; and two lidar sensors: HSRL-2 (Burton et al., 2015) and CPL (the Cloud Physics Lidar) (McGill et al., 2002), and flew over a variety of scenes at a high altitude approximately 20 km (Knobelspiesse et al., 2020). Several MAP aerosol retrievals from ACEPOL measurements have been reported (Fu et al., 2020; Puthukkudy et al., 2020; Gao et al., 2020; Hannadige et al., 2021; Gao et al., 2021a).

There are a total of five AirHARP ocean scenes available in ACEPOL. Three scenes on Oct 23, 2017 (Scenes 1, 2 and 3)

have been discussed by (Gao et al., 2021a, b). This study further analyzes the retrieval uncertainties on Scenes 2 and 3 and adds two additional scenes from Oct 27 (Scene 4) and Nov 07 (Scene 5). The adaptive data screening method of (Gao et al., 2021b) was applied on all these scenes to mask out viewing angles contaminated by cirrus clouds, ocean surface floating objects, or other irregularities that could not be represented adequately by the current forward model.

Fig. 10 shows retrieval results for Scene 2, with AOD and Rrs (both at 550 nm) in panels (b) and (c) and their retrieval

uncertainties shown in panels (e) and (f), respectively. The retrieved AOD and Rrs are reasonably smooth, varying mostly



in the ranges 0.07-0.1, and 0.003-0.004 respectively. Panel (d) shows the total number of observations used in the retrieval, which decreases toward the bottom of the image due to sunglint as shown in panel (a). Less number of measurements are also available at the top edge of the image due to the sensor geometry, which also results in larger AOD and $R_{rs}$ uncertainties. There are several patches elsewhere with fewer measurements due to the removal of cirrus cloud-contaminated angles (Gao et al., 2021b). Most pixels have at least 100 suitable measurements; the largest number of observations available is 228. Higher measurements are generally associated with lower uncertainties for both AOD and $R_{rs}$. Patches with small $R_{rs}$ values in the upper right portion of panel (c) also have larger uncertainties in panel (f). Retrieval uncertainties can be used as a flexible quality flag for each pixel, which is more effective than relying solely on the number of measurements or the cost function values only, as uncertainty estimates are specific to each retrieved parameter.

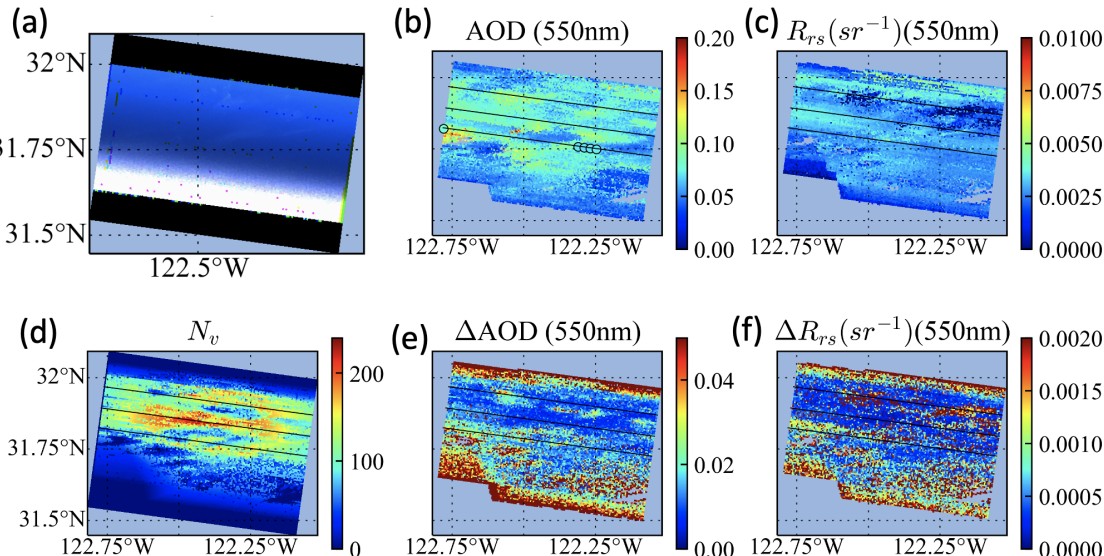

**Figure 10.** (a) RGB image for Scene 2 on Oct 23, 2017. Retrieval results are shown for (b) AOD and (c) $R_{rs}$, respectively; retrieval uncertainties for these are shown in (e) and (f), respectively. (d) the number of total observations used in retrievals. The HSRL AOD at 532nm are indicated at panel (b). More detailed analysis on AOD and its uncertainties over the three solid lines are discussed in Fig. 11.

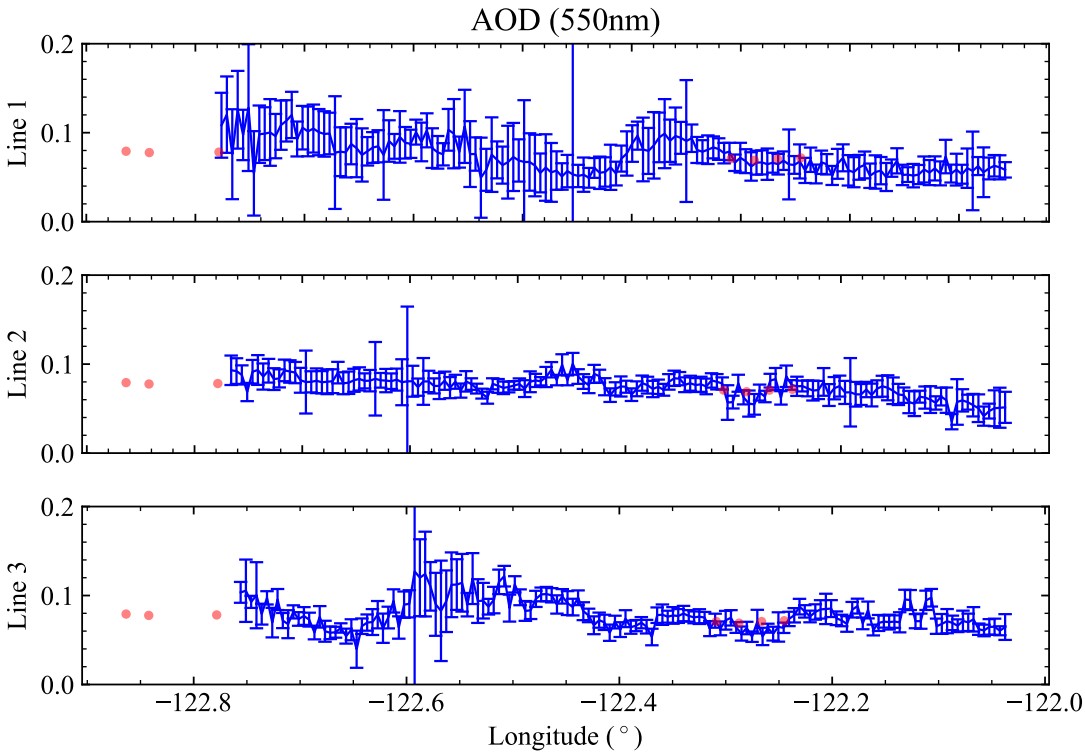

**Figure 11.** Retrieved AOD at 550 nm and their uncertainties along the three lines shown in Fig. 10 from bottom to top. Red dots are the HSRL AOD at 550 nm as indicated in Fig. 10

Fig. 11 shows the retrieved AOD at 550 nm and its uncertainties along the three black lines in Fig. 10(a). Line 1 contains the pixels closest to the HSRL track. Due to the impact of cirrus clouds, only a few HSRL pixels are available, but they agree with the retrieval results within the estimated uncertainties. The regions with cirrus cloud angles removed by the adaptive data screening approach also show larger uncertainties (the left portion of line 1 and the peak in line 3 near -122.6° longitude). The measurements in line 2 are less impacted by cirrus clouds with an average of 155 observations per retrieval, compared to 91

and 120 for lines 1 and 3 respectively. The $\chi^2$ map (shown in Gao et al. (2021b)) shows that excluding the cirrus-contaminated angles makes retrieval cost more uniform across the scene. The mean $\chi^2$ values along the three lines are 1.54, 1.25 and 1.34; since these $\chi^2$ are still larger than 1, there may be additional relevant uncertainties not captured in the input uncertainty model that require future investigation.



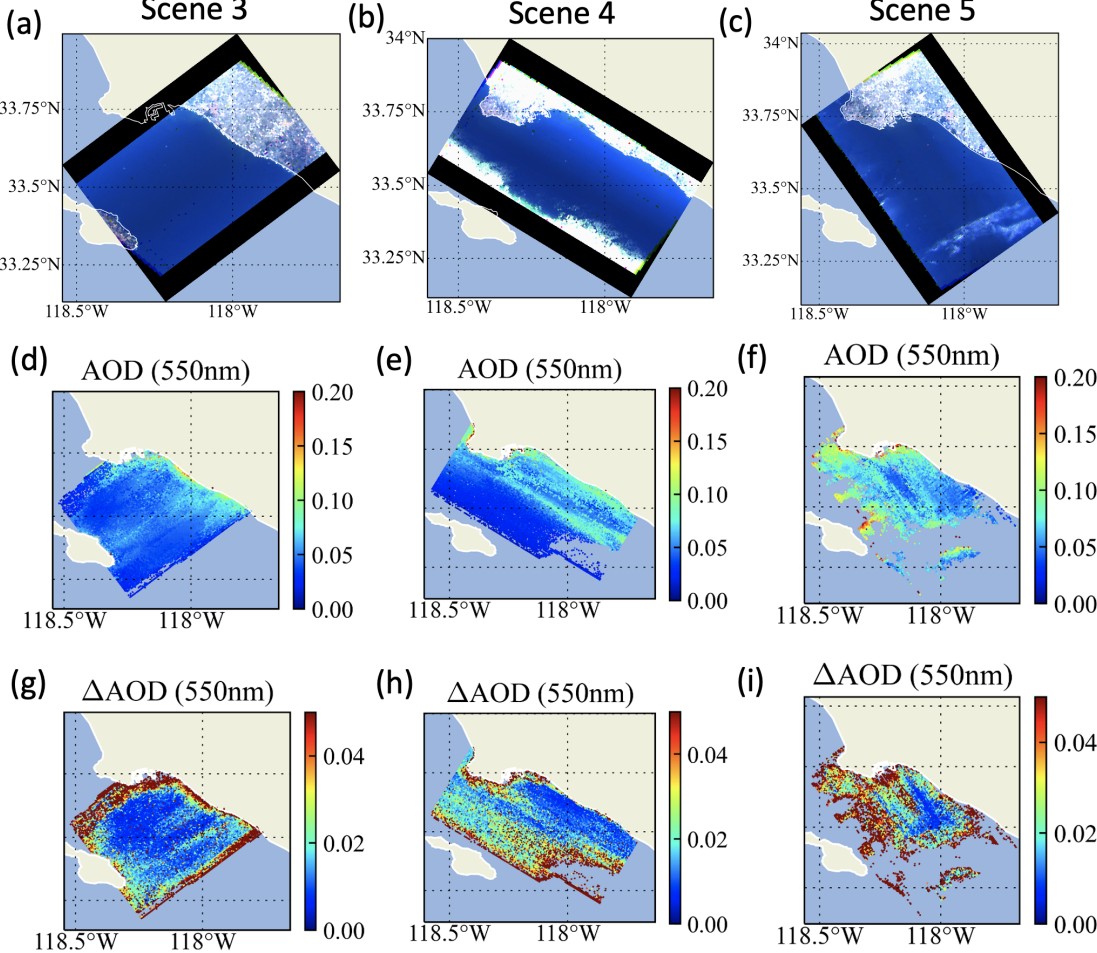

**Figure 12.** Three AirHARP scenes on Oct 23, Oct 27 and Nov 7, 2017, which in different flight directions but over the same region. The RBG images are shown in (a,b,c), the retrieved AOD at 550 nm are shown in (d,e,f), and their uncertainties are shown in (g, h, i).

Equivalent results for the other three Scenes(3,4,5) are shown in Fig. 12. The most probable $\chi^2$ are 1.2, 1.4 and 0.8 respec-
tively. For Scene 3, the retrieved AOD values are mostly around 0.05, but increase up to 0.1 near the coast as shown in Fig.
12(d). The retrieval uncertainties as shown in panel (h) are typically around 0.01, but exceed 0.05 near the coast and the edge
of the image. For retrievals uncertainties larger than 0.05, the average number of measurements is less than 22, but for those
with uncertainties under 0.05, an average of 80 measurements were available. Scene 4 is similar, although with sunglint in the
lower portion of image and larger associated uncertainties. For Scene 5 in Fig. 12, many pixels in the left and lower half of the
image are impacted by the cirrus clouds, often leaving few suitable angles and leading to AOD uncertainty larger than 0.05 (the
brown color shown in panel (i)). The central region with the smallest AOD uncertainties less than 0.01 corresponds to pixels
with 161 or more observations.





## 6 Discussions and conclusions

Quantifying the uncertainties associated with remote sensing retrievals is key to understanding retrieval performance, and gauging the quality and utility of the retrieval results. Retrieval uncertainties depend on the spectral, angular, radiometric, and polarimetric characteristics of the instrument. Increasing dimensionality and accuracy of measurements benefits retrievals but also introduces new challenges in the inversion of geophysical properties and estimation of retrieval uncertainties.

This study discussed and applied a practical, efficient way to estimate theoretical uncertainties for aerosol and ocean data products retrieved by FastMAPOL from synthetic AirHARP and HARP2 measurements, and field AirHARP measurements from the ACEPOL field campaign. Theoretical retrieval uncertainties for aerosol and ocean color properties are discussed. The speed with which the uncertainties can be computed is optimized using analytical derivatives based on automatic differentiations. To validate how well the retrieval uncertainties represent real retrievals, we provided a flexible Monte Carlo Error Propagation (MCEP) method to compare the retrieval uncertainties from error propagation with errors from synthetic retrievals. More discussions are as follows:

1. Using MCEP, statistical distributions can be compared to understand their properties and develop proper metrics for comparison. The real and theoretical retrieval uncertainties for multiple retrieval parameters are compared directly by their error histograms sampled from the Monte Carlo method based on the synthetic data retrievals. The ratios of the statistical metrics such as MAE for theoretical and real errors are computed and compared. These ratios provide a tool to quantify the overall performance of the retrieval uncertainty. The ratios are mostly 1-1.5 with respect to different AOD ranges which suggests that the FastMAPOL retrieval algorithm performs well as it approaches the optimal uncertainties predicted from error propagation. The larger ratios observed for aerosol refractive indices suggest a need to improve constraints on and/or test for proper convergence of those parameters, especially for cases with small AODs. Future studies of synthetic data with realistic statistics are needed to further evaluate the overall performance of the retrieval algorithm.

2. Synthetic data are only one piece of the evaluation and are limited because they use the same underlying forward model as the retrieval. Future comparison of retrieval results with in-situ measurements is desirable to provide a more complete assessment. However, what is available at present for AirHARP is sparse in volume, as AirHARP data are only available for a few field campaigns and PACE has not yet launched. Notably, there is no avenue to validate all retrieved products at once. The MCEP method and others (e.g. Hasekamp and Landgraf (2005); Sayer et al. (2020)) can also be used to compare uncertainty estimates with the in-situ measurements. Furthermore, the MCEP method provides a flexible framework to evaluate the uncertainties associated with the number of cases used in the statistical comparison, which can often be sparse for in-situ data. Use of in-situ data, however, also involves additional measurement and co-location uncertainties not included in the input uncertainty model (e.g. Virtanen et al. (2018); Sayer (2020)). Additionally, they may reveal assumptions in the forward model that are insufficient. For example, for coastal waters, we may need a more complicated ocean bio-optical model as demonstrated by Gao et al. (2019). The parameterization of aerosol size bins and refractive index spectral shape may also need refinement.





3. The Monte Carlo method has been used widely for uncertainty quantification due to its flexibility and robustness (e.g. Andrieu et al. (2003)). In this work, the theoretical retrieval uncertainties are still computed through the error propagation method. However, to validate the theoretical uncertainties, we need to compare with reference truth data, which is often

limited by its sample size, especially for in-situ measurements. It is important to consider the impacts of sample size and the statistical distribution on the robustness of metrics used in the analysis. In this study, we chose a Monte Carlo method to randomly sample errors from theoretical uncertainties, which provides a direct bridge to compare with the real retrieval errors. The current MCEP method generates random errors from the theoretical uncertainties derived through error propagation in step 2; another approach is to generate random errors directly from the error covariance matrix in

Eq. 4 and then propagate them through Eq. 6. The latter would be more flexible to deal with more general measurement uncertainty statistics, but more computationally expensive due to the large number of measurements present in MAP retrievals. Our MCEP method can be further developed to understand the impact of a priori constraints, broader statistical types of measurement errors, for better validation and understanding of retrieval uncertainties.

4. Retrieval initialization and convergence can be important. Gao et al. (2020) discussed the impact of initial values by

conducting hundreds of retrievals using random initial values and found the RMSE of the retrieval results produced a value similar to the error propagation results reported by Knobelspiesse et al. (2012). As discussed in Sec. 5, the cost function may not always converge to the values expected from $\chi^2$ distribution, and large values are often observed as shown by Wu et al. (2015); Gao et al. (2020, 2021a). This may be due to the impacts of anomalies not captured by the forward model (such as, here, cirrus clouds) or modeled but not quantified adequately in the input uncertainty model for

measurements plus forward models. Theoretical error propagation can give inaccurate results in these cases. It would be practical to remove such anomalous measurements from the retrieval, as in the adaptive data screening method by Gao et al. (2021b). Fewer suitable measurements tend to mean larger retrieval uncertainty, although this is arguably preferable (considering data coverage) to discard the whole retrieval based on a high-cost function. In these situations, the theoretical uncertainty estimate may guide whether a retrieval is useful for a particular application on a per-parameter

basis.

This work provides a general framework to understand the uncertainties from the retrieval algorithm and provides a bridge from theoretical uncertainty toward future evaluation using in-situ measurements. More complex input uncertainty model, such as the one including uncertainty correlations between the multi-angle measurements, can be evaluated based on this framework, which will be a topic in the Part 2 of this paper series. Although based on synthetic and airborne measurements, the methods

on uncertainty quantification are flexible and can be applied to existing and future satellite missions such as NASA's PACE mission with advanced multi-angle polarimetric instruments.



*Data availability.* The AirHARP and HSRL-2 data used in this study are available from the ACEPOL data portal (https://doi.org/10.5067/ SUBORBITAL/ACEPOL2017/DATA001, ACEPOL Science Team, 2017). The AirHARP L2 data product and their uncertainty files are available upon request from the corresponding author.

*Author contributions.* MG, KK, BF, P-WZ formulated the original concept. MG developed the algorithm and generate the scientific data. P-WZ developed the radiative transfer code used in the simulations. KK, AS, AI, YH and OH advised on the uncertainty models. KK, P-WZ, AS, BC and OH advised on the aerosol products. BF, AI and JW advised on the ocean color products. VM and XX provided and advised on the HARP data. MG wrote the manuscript draft. All authors provided critical feedback and edited the manuscript.

*Competing interests.* The authors declare no conflict of interest.

*Acknowledgements.* The authors would like to thank the ACEPOL teams for conducting the field campaign, thank the HARP and HSRL teams and PIs for providing the data, and thank the NASA Ocean Biology Processing Group (OBPG) system team for supporting the High Performance Computing (HPC).

    MG, KK, BF, AS, AI, BC, JW have been supported by the NASA PACE project. P-WZ and YH have been supported by NASA (grants 80NSSC20M0227). The ACEPOL campaign has been supported by the NASA Radiation Sciences Program, with funding from NASA (ACE 
and CALIPSO missions) and SRON. Part of this work has been funded by the NWO/NSO project ACEPOL (project no. ALWGO/16-09).



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
