# Peer review of "Effective uncertainty quantification for multi-angle polarimetric aerosol remote sensing over ocean"

_Atmospheric Measurement Techniques, 2022_

## Referee Comment (RC3)

Review of manuscript submitted to AMT:

Title: Effective uncertainty quantification for multi-angle polarimetric aerosol remote sensing over ocean, Part 1: performance evaluation and speed improvement

Authors: Meng Gao et al.

**General Comments**

In this paper, the authors analyze theoretical uncertainty estimates and validate them using a Monte Carlo approach to generate error statistics. A previously developed Fast Multi-Angle Polarimetric (MAP) Ocean coLor (FastMAPOL) retrieval algorithm is used to carry out the retrievals and quantify uncertainties for both synthetic HARP2 (Hyper-Angular Rainbow Polarimeter 2) and AirHARP (airborne version of HARP2) datasets. The FastMAPOL retrieval algorithm is based on neural network (NN) forward vector radiative transfer model (VRTM) simulations pertinent for a coupled atmosphere-ocean system. The NN forward radiative transfer models are trained using synthetic data generated by the VRTM. For practical application of the appraoch to uncertainty evaluation in operational data processing, the authors apply a previously developed automatic differentiation method to calculate derivatives (Jacobians) analytically based on the neural network models.

Both the speed and accuracy associated with the quantification of uncertainties for retrievals based on MAP data are presented and discussed. Pixel-by-pixel retrieval uncertainties are evaluated for synthetic data as well as data obtained in AirHARP field campaigns.

The authors argue that the methods and results presented in this paper can be used to evaluate the quality of data products, and guide algorithm development based on MAP measurements for current and future satellite systems.

The paper is generally well written and the results appear to be robust and valuable. Therefore, I recommend that the paper be published after minor revisions outlined below.

**Specific Comments**

- On line 6, the authors mention "nonlinearity of radiative transfer near the solution". Since the VRTE is a linear equation, the authors should clarify what they mean here and use proper wording.

- line 53: the statement "the forward model is linear near the solution" also needs rewording, because the RTE is a linear equation.

- line 54: change "With MAPs, theoretical uncertainties...." to something like "For MAP measurements, theoretical uncertainties...."

- Line 62: change "error propagation does but require" to error propagation but requires"

- line 64: change "With MAPs, theoretical uncertainties...." to something like "For MAP measurements, theoretical uncertainties...."

- Line 74: change "Several approaches has been proposed" to "Several approaches have been proposed"

- Line 78: explain what "non-linearity around the truth" is supposed to mean

- change "properties retrieved directly by the MAP" to "properties retrieved directly from the MAP data"

- "more advanced instruments" please be more specific

- Line 114 "Section 4. quantified" → 'Section 4 quantifies"

- Line 115 "Section 5. quantified" → 'Section 5 quantifies"

- Line 125: "There are two MAPs on PACE" → "There are two MAP instruments on PACE"

- Line 147: "lower-dimensionality retrievals" – please explain.

- Line 158: "assumed as a combination of five lognormally-distributed aerosol sub-modes" – please justify this choice

- Line 175: "the spectral ocean color remote sensing reflectance $(R_{rs}(\lambda))$ is derived based on the retrieved aerosol properties through atmospheric correction" – a physically more satisfactory and accurate approach is presented by Fan et al. (2021).

- Line 192: "STIR is based on the Levenberg-Marquardt algorithm combined with ..... " – Please summarize the advantage of the STIR method compared to a "standard" Levenberg-Marquardt algorithm.

- Equation (14) – a physically more satisfactory and accurate approach consistent with the coupled atmosphere-ocean system is provided by Fan et al. (2021).

- Line 354: "Note that the synthetic data is computed directly using the vector radiative transfer model, but the NN forward model is used in the retrieval algorithm." – Please explain the significance/advantage of this approach.

- Line 415: "retrieval algorithms can be further improved, for instance, by including additional a priori constraints" – what kind of constraints? – please be more specific.

- "Less number of measurements are" → "A smaller number of measurements is"

- Line 440: "Higher measurements are generally" → '' A larger number of measurements is generally"

- Line 450: "makes retrieval cost more uniform" – not clear, please rewrite.

- 495: "a more complicated ocean bio-optical model" $\rightarrow$ "a more complete and realistic ocean bio-optical model"

**Technical Corrections**

In general this paper is well written, and I did not spot any typographical or grammar mistakes, except for the following:

- Line 204: "explicit a prior information" $\rightarrow$ "explicit *a priori* information"

- Line 206: "we assume Sa" $\rightarrow$ "we assume $\mathbf{S}_a$"

- Line 207: "assumed a prior " $\rightarrow$ "assumed *a priori*"

- Line 329: "Rrs" $\rightarrow$ "$R_{rs}$"

- Line 337: "their difference are quantified" $\rightarrow$ "their difference is quantified"

- Line 405: "errors are found sufficiently to evaluate" $\rightarrow$ 'errors are found sufficient to evaluate"

- Line 518: "based on a high-cost function" $\rightarrow$ "based on a failure to properly minimize the cost function"

**Reference**

Fan, Y., W. Li, N. Chen, J.-H. Ahn, Y.-J. Park, S. Kratzer, T. Schroeder, J. Ishizaka, R. Chang, and K. Stamnes, OC-SMART: A machine learning based data analysis platform for satellite ocean color sensors, Remote Sensing of the Environment, 253, 112236, 2021.

---

## Author Comment (AC1)

**RC1**: 'Comment on amt-2022-112', Feng Xu, 06 Jun 2022

The paper by Gao et al. validates a Bayesian uncertainty propagation model against the \*real\* uncertainty estimate from analyzing synthetic retrievals. The simulation study indicates that the theoretical uncertainties of the retrieved pixel aerosol quantities basically reproduce the real retrieval uncertainties in most cases, though with certain degree of underestimates. This is a major finding and should be useful for aerosol and ocean color remote sensing using PACE polarimeters. Another valuable observation is about the importance of sample size for uncertainty estimate. The authors shows the increased robustness of their uncertainty analysis when the sample size increase from 50 to 1000. Following the validation, retrievals were performed using the real observations from AirHARP field measurements and the Bayesian uncertainty estimate model. In algorithm development, the authors deployed the FastMAPOL approach, which couples NN based RT calculation and the automatic difference method for Jacobian evaluation. These two elements ensure the efficiency of the uncertainty model assessment in the present work.

We appreciate your time and efforts in reviewing this work. The comments and questions are valuable in improving the clarity of this manuscript. The responses are below (in red) with manuscript revised accordingly (also uploaded).

Overall, the paper was well-written. I have the following four comments for the authors to consider to clarify their approach:

Thank you for the positive comments on our work.

1. Do I understand correctly that the retrieval results for statistical analysis (e.g. those in Figs. 5-6) subjected to certain the convergence criterion ? For example, the cost function needs to be less than or equal to the metric unit when the retrieval is flagged to be successful so that the results are further used in the uncertainty analysis. Associated with this question, what is the success rate of the retrieval ?

   The reviewer is correct. We have chosen a maximum cost function value of 1.5 for the analysis of the synthetic data retrievals. However, this corresponds to a very high success rate of 96% for AirHARP and 93% for HARP2. In this way, only outliers are removed, and the main statistical distribution are maintained in the analysis. The cost function histogram is shown in Fig. 3, also copied below:

[Figure]

Figure 3. The histogram of the cost function values for the synthetic retrievals.

We added the following in our manuscript:
"

**To reduce the impact of outliers, we choose a maximum $\chi^2$ value of 1.5 in this study as shown in Fig. 3, which corresponds to a success rate of 96% for AirHARP cases and 93% for HARP2 cases.** "

2. Table 1 is commendable as it lists the range of 11 retrieval parameters which further decides the a priori matrix used in retrieval. Could the authors comment on whether there is potential impact of the a priori on the conclusion ? For example, if we relax the upper bound of the imaginary part of refractive index to be larger (e.g. 0.1 or larger for some strongly absorbing aerosols), will the Bayesian model based uncertainty still mimic the real uncertainties, or there might be additional underestimates of uncertainty ?

This is an excellent question and will be a very interesting topic for future study.

When increasing the bound of the retrieval parameters, it is likely to have more challenges to correctly retrieve the corresponding parameter. 1) This may relate to the accuracy of the forward model and the neural network model used to represent the forward model. For example, more sampling point may be required to generate the neural network training data, to accurately represent the forward model, and its Jacobians. 2) There may be also impact of the local minima of the cost function around the new territory of the parameter. Therefore, it is important to quantify the difference between the theoretical uncertainty and the real uncertainty when new parameter range are used. Although, our current study only focuses on weakly absorbing aerosols (with imaginary refractive index <0.03), but the approach proposed in this study can be a useful tool to access such impacts in a future study when more complex aerosols are presented.

We added the following discussion in the manuscript in Sec 2.2:

**"In this work, we only consider weakly absorbing aerosols with mi<0.03. It will be a subject of future studies on how the theoretical uncertainties represents the real uncertainties for more complex aerosol models, following the approach discussed in this study.**
"

3. Eq.(5): Is the modeling error (e.g. five size components of aerosols, Cox-Munk ocean surface, etc) excluded or included in the piece of VRTM model uncertainty "sigma_{VRTM}" ? I'm curious in this aspect since in real data retrieval (e.g. the demonstrated AirHARP retrieval), one of the major error sources is the modeling errors. To enhance the connection of synthetic retrieval and AirHARP retrievals, it would be great if modeling error is included in the Bayesian model via RT simulation uncertainty "sigma_{VRTM}" and then via Eqs. (5)-(6).

Thank the reviewer for the interesting question which is important for the application to real measurement. We do not consider explicit modeling errors in the total uncertainty model in this work, but we have taken efforts to reduce the impact of modeling uncertainties.

For the use of Cox-Munk ocean surface model, we do observe discrepancy in fitting the sunglint signal, which may relate to wind direction or ocean swell. Therefore, we have removed the sunglint signal within 40 degree with respect to the spectral reflection direction to minimize its impacts. The following discussion is included in the manuscript:

> "Strong sunglint is excluded here by removing view angles within 40º_of the specular reflection direction due to the challenges to represent the sunglint signals from ACEPOL field campaign using the isotropic Cox and Munk model (Gao et al 2020, Gao et al 2021a)."

For the use of five size aerosol model, it provides a robust approach to retrieve aerosol size parameters, which have also been demonstrated by Dubovik et al., 2006 and Xu et al., 2016. Moreover, for a more general study, Fu and Hasekamp discussed the representation of aerosol size distribution through various numbers of sub-modes and also found that a similar five-mode approach can provide good retrievals for most aerosol parameters (Fu and Hasekamp, 2018). We have revised our discussed as follows:

> "The aerosol size distribution is assumed as a combination of five lognormally-distributed aerosol sub-modes, each with prescribed mean radii and variances; the five volume densities ($V_i$) are free parameters (Dubovik et al., 2006; Xu et al., 2016). **The five-mode approach is found to provide good retrievals for most aerosol parameters (Fu and Hasekamp, 2018).**

Moreover, the modeling error is very challenging to quantify, since it depends on the model itself used in the RT simulation, and the real measurements selected for retrieval, which can vary pixel by pixel. For example, when there is contamination of the thin cirrus cloud in real data, the aerosol only model will correspond to a large uncertainty (even "wrong" results when the cirrus cloud signal is strong). We have been using a data screening approach to reduce the discrepancy between the model and data used in retrievals, and we also expect the uncertainty quantification

approach proposed in this study can be also used to provide a tool in understanding and accessing modeling errors. Here are more discussions:

1) To minimize the impact of the scenarios with insufficient models in this study, we are using the data screening approach developed in Gao et al 2021b, Frontiers. The measurement which cannot be fitted well by the forward model are removed based on the ratio of fitting residual and an uncertainty model. This approach is applied adaptively by conducting retrieval several times. Through this approach, the forward model is more likely to be applied on the measurement proper for its design and therefore reduce the modeling error.
2) The uncertainty quantification approach itself can be also used to analyze the modeling error. We can analyze the retrieval residuals after data screening, and compare its statistics directly with the assumed uncertainty model. If we find there are differences, it is likely due to the impact of modeling error.

We added the following sentence in Sec 2.3

**"As discussed in Sec. 1, an adaptive data screening method is used to remove the real measurements which cannot be fitted well by the forward model (Gao et al., 2021b). In this way, the impact of forward model uncertainties can be reduced. We do not consider additional forward model uncertainties in this study.
"**

4. What is convergence metric for AirHARP's real data retrieval ? Is it consistent with the those used in the synthetic retrievals ?

For the study of real data, we consider all the retrieval cases. This is because the data screening approach has been applied on the real data retrievals, which removed the measurement cannot be fitted well by the forward model. The maximum cost function is often less than 3. Therefore, we do not define a maximum cost function to remove the outliers. For the measurement cannot be fitted by the forward model at all, the number of valid angle Nv will be zero, such as in Fig 10, panel d, some pixels in the center of the image are removed completely.

As an example, the cost function histogram for scene2 before and after data screening have been shown in Gao et al 2021, Frontiers paper as copied below. The red histogram is after data screening. Note that the center of the histogram is slightly larger than 1 which may relate to the contribution of modeling error.

[Figure]

We revised the following discussion in the manuscript:

"The adaptive data screening method of (Gao et al., 2021b) was applied on all these scenes to mask out viewing angles contaminated by cirrus clouds, ocean surface floating objects, or other irregularities that could not be represented adequately by the current forward model. **The resulted cost function histogram is much better described by the $\chi^2$ distribution using the assumed uncertainty model (Gao et al., 2021b). "**

---

## Author Comment (AC2)

**RC2: 'Review', Anonymous Referee #2, 15 Jun 2022**

The Authors discuss a method to evaluate theoretical uncertainties in retrieved aerosol and ocean surface properties from multi-angle polarimetric remote sensing measurements. Through a Monte Carlo sampling approach, they propose a method to compare theoretical uncertainties with observed observed-true parameter differences computed from validation. The method is applied to synthetic as well as to real AirHarp measurements. In addition, the Authors briefly discuss a way to speed up a posteriori uncertainty calculations by analytic differentiation of a neural network-based forward model.

The topic is very interesting and within the scope of AMT, the analyses appear sound and convincing. The only slight criticism that I have is that I found the paper rather long and not always easy to follow. A large amount of detailed analyses are presented, and the task of deciding what are the most important points of the study appears to be mostly left to the reader. I wonder if the information can be synthesised a bit to make the paper easier to read. Furthermore, it seems to me that the aspects of "performance evaluation" and "speed improvement" are a bit intermixed when it comes to the way the sections of the manuscript are organized, which does not help readability. Given that it seems to me that the "performance evaluation" aspect of the manuscript is given a much wider space than the "speed improvement",

We appreciate your time and efforts in reviewing this work. The comments and questions are valuable in improving the clarity of this manuscript. The responses are below (in red) with manuscript revised accordingly (also uploaded).

I wonder whether it would be better to move this latter to an appendix. Apart from this, I think this is an excellent paper. Below are a few other minor comments:

Thank the reviewer for the positive comments. Following the reviewer's suggestion, we have moved section 3 to Appendix with details provided below.

- L113: A more recent reference is

Fougnie, B. et al. (2018), "The multi-viewing multi-channel multi-polarisation imager – Overview of the 3MI polarimetric mission for aerosol and cloud characterization", JQSRT, 23-32, doi: 10.1016/j.jqsrt.2018.07.008

Thank you for suggesting the reference. It has been updated.

- L156-157. Does your forward model covers the entire HARP spectral range?

If so, is it not unrealistic to assume spectrally flat refractive index?

Especially dust aerosols are way more absorbing in the UV than in the VIS/NIR.

For HARP/AirHARP, there are four bands: 440, 550, 670 and 870nm. The refractive index spectral are quite flat within this spectral range as we have been used in our previous study (Gao et al, 2021, AMT). However, UV channels are available for OCI and SPEXone sensors on PACE, which will require a more sophisticated refractive index spectra as the reviewer pointed out.

- L173-175. "The forward calculation of aerosol size etc.".

I suggest rephrasing as... "The forward calculation of aerosol optical depth (AOD)

and single scattering albedo (SSA) from aerosol size and refractive index"

Thank you for the suggestion. We have revised the sentence as

"The forward calculation of aerosol optical depth (AOD) and single scattering albedo (SSA) from aerosol size and refractive index is also performed using NNs based on simulations using the numerical code based on the Lorenz-Mie theory (Mishchenko et al 2002)"

- L192. "interior method". Do you mean "interior point method"?

Our use of "interior method" is to follow the naming convention in the algorithm paper (Branch et al 1999). The algorithm uses the 'affine scaling' which ensure that the solution is searched strictly within the interior of the feasible region. It seems "interior point method" is often used to refer the algorithm with a barrier function.

We also added more information about the STIR method as follows:

"The subspace trust-region interior reflective (STIR) algorithm is employed to conduct non-linear least-square minimization of the cost function (Branch et al., 1999). **Its implementation in the Python package SciPy is used in this study (Virtanen et al., 2020).** STIR is based on the Levenberg-Marquardt algorithm combined with an interior method and reflective boundary technique (Branch et al., 1999). **The interior method ensure that the retrieval parameters are searched strictly within the interior of the feasible region as specified in Table 1, while the reflection technique can significantly reduce the number of iterations in the minimization process.**"

- Personally I think section 3 breaks the flow of the paper a bit. While the paper

is mostly about illustrating how theoretical uncertainties compare with observed errors,

here you discuss a technical detail of how to speed up theoretical uncertainty calculations.

Wouldn't it be better to have this as an appendix?

We have further synthesized the manuscript by moving the "speed improvement" section to the appendix A. At the end of Sec 3.1, we added a sentence on the use of automatic differentiation, and leave all the details in Appendix:

> **"Automatic differentiation is used to calculate both the Jacobian matrix and the derivatives defined in Eq. 8 for AOD and SSA, as well as water leaving signals involving atmospheric correction and BRDF correction. More details are discussed in Appendix A."**

Please note that the title is also updated:

> "Effective uncertainty quantification for multi-angle polarimetric aerosol remote sensing over ocean"

- L413. A ratio of 1.5 means a 50% difference. Not sure I would regard this as a "slight underestimate"

Thank you for pointing this out. We have revised the sentence as follows

> "The ratios are mostly in the range 1-1.5, indicating that the theoretical uncertainties work well to represent the real retrieval uncertainties in most cases but are generally underestimates."

---

## Author Comment (AC3)

**RC3: 'Comment on amt-2022-112', Anonymous Referee #3**

Review of manuscript submitted to AMT:
Title: Effective uncertainty quantification for multi-angle polarimetric aerosol remote sens-ing over ocean, Part 1: performance evaluation and speed improvement
Authors: Meng Gao et al.

General Comments
In this paper, the authors analyze theoretical uncertainty estimates and validate them using a Monte Carlo approach to generate error statistics. A previously developed Fast Multi-Angle Polarimetric (MAP) Ocean coLor (FastMAPOL) retrieval algorithm is used to carry out the retrievals and quantify uncertainties for both synthetic HARP2 (Hyper-Angular Rainbow Polarimeter 2) and AirHARP (airborne version of HARP2) datasets. The FastMAPOL retrieval algorithm is based on neural network (NN) forward vector radiative transfer model (VRTM) simulations pertinent for a coupled atmosphere-ocean system. The NN forward radiative transfer models are trained using synthetic data generated by the VRTM. For practical application of the appraoch to uncertainty evaluation in operational data processing, the authors apply a previously developed automatic differentiation method to calculate derivatives (Jacobians) analytically based on the neural network models. Both the speed and accuracy associated with the quantification of uncertainties for retrievals based on MAP data are presented and discussed. Pixel-by-pixel retrieval uncertainties are evaluated for synthetic data as well as data obtained in AirHARP field campaigns. The authors argue that the methods and results presented in this paper can be used to evaluate the quality of data products, and guide algorithm development based on MAP measurements for current and future satellite systems.

We appreciate your time and efforts in reviewing this work. The comments and questions are valuable in improving the clarity of this manuscript. The responses are below (in red) with manuscript revised accordingly (also uploaded).

The paper is generally well written and the results appear to be robust and valuable. Therefore, I recommend that the paper be published after minor revisions outlined below.

Thank the reviewer for the positive comments.

Specific Comments
• On line 6, the authors mention "nonlinearity of radiative transfer near the solution".
Since the VRTE is a linear equation, the authors should clarify what they mean here and use proper wording.

Thank you for the suggestion. Here the nonlinearity is referring to the non-linear relationships between the total reflectance and polarization with respect to the retrieval parameters. We revised the sentence as follows:

> "However, standard error propagation techniques in high-dimensional retrievals may not always represent true retrieval errors well due to issues such as local minima and **the**

**nonlinear dependence of the total reflectance and degree of linear polarization (DoLP) on the retrieved parameters near the solution."**

• line 53: the statement "the forward model is linear near the solution" also needs rewording, because the RTE is a linear equation.

Here we are discussing the assumptions in error propagation. Similar to above discussions, we revised the sentence as follows:

"…a) the input uncertainty model is sufficient, b) the retrievals converge to their global minimum, c) **the forward model is linear with respect to the retrieval parameters near the solution.**
"

• line 54: change "With MAPs, theoretical uncertainties...." to something like "For MAP measurements, theoretical uncertainties...."
Thanks for the suggestions. We have revised the sentence as:
"**For MAP measurements,** theoretical uncertainties have been widely used for aerosol and cloud retrieval algorithms for sensors…"

• Line 62: change "error propagation does but require" to error propagation but requires"
We have revised the sentence as:
"The 'real' uncertainty does not require the same assumptions as error propagation but **requires** the existence of 'truth' data of high and known confidence,…"

• line 64: change "With MAPs, theoretical uncertainties...." to something like "For MAP measurements, theoretical uncertainties...."
We have revised the sentence as:
"**For MAP measurements,** real uncertainties have been discussed for aerosols over ocean, land, and cloud by comparing retrievals with synthetic data and in-situ measurements…"

• Line 74: change "Several approaches has been proposed" to "Several approaches have been proposed"
Updated.

• Line 78: explain what "non-linearity around the truth" is supposed to mean
We have discussed the non-linearity in the response to the first two comments.

• change "properties retrieved directly by the MAP" to "properties retrieved directly from the MAP data"

Updated

• "more advanced instruments" please be more specific
We have revised the sentence as below:

"…and can guide further uncertainty studies and algorithm development when more advanced instruments **with higher angular and spectral resolutions** are available."

• Line 114 "Section 4. quantified" → 'Section 4 quantifies"
Updated

• Line 115 "Section 5. quantified" → 'Section 5 quantifies"
Updated

• Line 125: "There are two MAPs on PACE" → "There are two MAP instruments on PACE"
Updated

• Line 147: "lower-dimensionality retrievals" – please explain.
We have revised the term as:
"… as is common for **retrievals with a small number of parameters**,…"

• Line 158: "assumed as a combination of five lognormally-distributed aerosol sub-modes" – please justify this choice

From our previous test with respect to the RSP data, we found the retrieval of the number density of each sub-mode is more robust than directly retrieve the effective size and variance of the aerosol volume distribution. For a more general study, Fu and Hasekamp discussed the representation of aerosol size distribution through various numbers of sub-modes and also found that a similar five-mode approach can provide good retrievals for most aerosol parameters (Fu and Hasekamp, 2018). We have added a sentence to reflect this:

> "The aerosol size distribution is assumed as a combination of five lognormally-distributed aerosol sub-modes, each with prescribed mean radii and variances; the five volume densities (Vi) are free parameters (Dubovik et al., 2006; Xu et al., 2016). **The five-mode approach is found to provide good retrievals for most aerosol parameters** (Fu and Hasekamp, 2018)**.**

• Line 175: "the spectral ocean color remote sensing reflectance (Rrs(λ)) is derived based on the retrieved aerosol properties through atmospheric correction" – a physically more satisfactory and accurate approach is presented by Fan et al. (2021).
Thank you for the reference. We have added it in the manuscript. We also provide more information on why we prefer the atmospheric correction approach. It will be useful to apply the MAP retrieved aerosol parameters to conduct atmospheric correction on other sensors, such as PACE OCI

"In addition, the spectral ocean color remote sensing reflectance (Rrs) is derived based on the retrieved aerosol properties through atmospheric correction, a procedure to derive ocean color signals by removing the contributions with atmosphere and ocean surface from the top of atmosphere (TOA) measurements (Mobley et al., 2016, 2022). **The atmospheric correction and other associated procedures have been implemented using NNs by Gao et al. (2021) with more details provided in Appendix A. The atmospheric correction method also provides a convenient way to derive ocean color signals from other sensors, such as PACE OCI, using the MAP retrieved aerosol properties. Note that NN method has also been used to directly link Rayleigh-corrected TOA radiances with normalized remote sensing reflectance by Fan et al (2021).**"

• Line 192: "STIR is based on the Levenberg-Marquardt algorithm combined with ....." – Please summarize the advantage of the STIR method compared to a "standard" Levenberg-Marquardt algorithm.
We have added more discussion as follows:

"The subspace trust-region interior reflective (STIR) algorithm is employed to conduct non-linear least-square minimization of the cost function (Branch et al., 1999). **Its implementation in the Python package SciPy is used in this study (Virtanen et al., 2020).** STIR is based on the Levenberg-Marquardt algorithm combined with an interior method and reflective boundary technique (Branch et al., 1999). **The interior method ensure that the retrieval parameters are searched strictly within the interior of the feasible region as specified in Table 1, while the reflection technique can significantly reduce the number of iterations in the minimization process.**"

• Equation (14) – a physically more satisfactory and accurate approach consistent with the coupled atmosphere-ocean system is provided by Fan et al. (2021).
We have added the reference and discussed the reason why we are conducting such atmospheric correction in the response to one of the previous comments. Please also note that the section containing the equation has been moved to Appendix A.

• Line 354: "Note that the synthetic data is computed directly using the vector radiative transfer model, but the NN forward model is used in the retrieval algorithm." – Please explain the significance/advantage of this approach.

We prefer to use the synthetic data directly computed by the radiative transfer model to capture any difference between the radiative transfer simulation and the NN models. As shown in Eq (5), the contribution of the NN uncertainties has been considered in the total uncertainty model. We have added the following discussions:

**"The synthetic data is computed directly using the vector radiative transfer model, but the NN forward model is used in the retrieval algorithm to achieve maximum efficiency. In this way the contribution of the NN uncertainties is captured both in the simulation and the uncertainty model as shown in Eq. 5. "**

• Line 415: "retrieval algorithms can be further improved, for instance, by including additional a priori constraints" – what kind of constraints? – please be more specific.

Multiple constraints can be added as summarized in Dubovik et al 2021. For example, there could be smoothness constraints on the refractive index spectra, volume distribution, vertical profile etc. We have revised the sentence as:
> "The large ratios of real and theoretical uncertainties also indicate where retrieval algorithms can be further improved, for instance, by including additional a priori constraints, **such as smoothness in refractive index spectra and size distribution, temporal and spatial variations of the retrieval parameters. Various constraints in the framework of multi-term least square method are summarized by Dubovik et al., (2021)."**

• "Less number of measurements are" → "A smaller number of measurements is"
Updated.

• Line 440: "Higher measurements are generally" → '' A larger number of measurements is generally"
Updated.

• Line 450: "makes retrieval cost more uniform" – not clear, please rewrite.
Since the cirrus cloud angles are removed, the cost function appear more uniform across the scene. We have revised the sentence as follows
> "The $\chi^2$ map (shown in Gao et al 2021) shows that excluding the cirrus-contaminated angles makes retrieval **cost function more spatially uniform** across the scene."

• 495: "a more complicated ocean bio-optical model" → "a more complete and realistic ocean bio-optical model"
Updated

Technical Corrections
In general this paper is well written, and I did not spot any typographical or grammar mistakes, except for the following:
• Line 204: "explicit a prior information" → "explicit a priori information"
Updated

• Line 206: "we assume Sa" → "we assume **S**a"
Updated

• Line 207: "assumed a prior " → "assumed a priori"
Updated

• Line 329: "Rrs" → "*Rrs*"
Updated, also checked the whole document.

• Line 337: "their difference are quantified" → "their difference is quantified"
Updated

• Line 405: "errors are found sufficiently to evaluate" → 'errors are found sufficient to evaluate"
Updated

• Line 518: "based on a high-cost function" → "based on a failure to properly minimize the cost function"
Thank you for the suggestion. We revised the statement slightly as follows:

> "Fewer suitable measurements tend to lead to larger retrieval uncertainty, although this is arguably preferable (considering data coverage) to **discarding the whole retrieval based on a high-cost function value.**"

Reference
Fan, Y., W. Li, N. Chen, J.-H. Ahn, Y.-J. Park, S. Kratzer, T. Schroeder, J. Ishizaka, R. Chang, and K. Stamnes, OC-SMART: A machine learning based data analysis platform for satellite ocean color sensors, Remote Sensing of the Environment, 253, 112236, 2021.